# Is it possible to compare inhibitory and excitatory intracortical circuits in face and hand primary motor cortex?

Francesca Ginatempo[1], Nicola Loi[1], Andrea Manca[1], John C. Rothwell[2] and Franca Deriu[1,3] 

[1] *Department of Biomedical Sciences, University of Sassari, Sassari, Italy*
[2] *Sobell Department of Motor Neuroscience and Movement Disorders, UCL Institute of Neurology, London, UK*
[3] *Unit of Endocrinology, Nutritional and Metabolic Disorders, AOU Sassari, Sassari, Italy*

Handling Editors: Richard Carson & Dario Farina

The peer review history is available in the Supporting Information section of this article (https://doi.org/10.1113/JP283137#support-information-section).

**Abstract**  Face muscles are important in a variety of different functions, such as feeding, speech and communication of non-verbal affective states, which require quite different patterns of activity from those of a typical hand muscle. We ask whether there are differences in their neurophysiological control that might reflect this. Fifteen healthy individuals were studied. Standard single- and paired-pulse transcranial magnetic stimulation (TMS) methods were used to compare intracortical

The Journal of Physiology

inhibitory (short interval intracortical inhibition (SICI); cortical silent period (CSP)) and excitatory circuitries (short interval intracortical facilitation (SICF)) in two typical muscles, the depressor anguli oris (DAO), a face muscle, and the first dorsal interosseous (FDI), a hand muscle. TMS threshold was higher in DAO than in FDI. Over a range of intensities, resting SICF was not different between DAO and FDI, while during muscle activation SICF was stronger in FDI than in DAO ($P = 0.012$). At rest, SICI was stronger in FDI than in DAO ($P = 0.038$) but during muscle contraction, SICI was weaker in FDI than in DAO ($P = 0.034$). We argue that although many of the difference in response to the TMS protocols could result from the difference in thresholds, some, such as the reduction of resting SICI in DAO, may reflect fundamental differences in the physiology of the two muscle groups.

(Received 23 March 2022; accepted after revision 13 June 2022; first published online 8 July 2022)
**Corresponding author** F. Deriu: Department of Biomedical Sciences, University of Sassari, Viale San Pietro 43/b, 07100 Sassari, Italy.     Email: deriuf@uniss.it

**Abstract figure legend** During muscle activation, in the hand primary motor cortex (M1) short-latency inhibitory and excitatory intracortical circuits (short interval intracortical inhibition (SICI) and short interval intracortical facilitation (SICF), respectively) are balanced, while in face M1 SICI is stronger than SICF. The strong inhibitory control of face M1 during voluntary muscle activation may facilitate access to face muscles' control from other cortical areas, such as those involved in the emotional control of these muscles. Image made using BioRender.com.

## Key points

- Transcranial magnetic stimulation (TMS) single- and paired-pulse protocols were used to investigate and compare the activity of facilitatory and inhibitory intracortical circuits in a face (depressor anguli oris; DAO) and hand (first dorsal interosseous; FDI) muscles. Several TMS intensities and interstimulus intervals were tested with the target muscles at rest and when voluntarily activated.
- At rest, intracortical inhibitory activity was stronger in FDI than in DAO. In contrast, during muscle contraction inhibitory activity was stronger in DAO than in FDI. As many previous reports have found, the motor evoked potential threshold was higher in DAO than in FDI.
- Although many of the differences in response to the TMS protocols could result from the difference in thresholds, some, such as the reduction of resting short interval intracortical inhibition in DAO, may reflect fundamental differences in the physiology of the two muscle groups.

## Introduction

Non-invasive brain stimulation has extensively contributed to understanding the physiological behaviour of different cortical regions (Rossini et al., 2015). A large body of the literature has shown that transcranial magnetic stimulation (TMS) paired-pulse protocols are useful tools to investigate the inhibitory and facilitatory circuitry of the human primary motor cortex (M1) (Kobayashi & Pascual-Leone, 2003). In particular, if a subthreshold conditioning pulse (CS) is applied through the same coil 1.0–5.0 ms before a suprathreshold test stimulus (TS), it is possible to elicit a short-interval intracortical inhibition (SICI) (Kujirai et al., 1993). SICI has

**Francesca Ginatempo** obtained her PhD in Neuroscience at the University of Sassari. In her PhD project she investigated the coordination of the facial muscles, and the influence of emotional stimulus on the corticobulbar motor control. She is now a Research fellow at University of Sassari. She is interested in the physiology and pathophysiology of the face motor system.

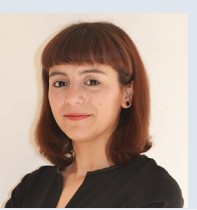

been largely characterized in both healthy subjects and neurological patients (Rossini et al., 2015). It has been shown that the interaction between conditioning and test pulses occurs at cortical level (Di Lazzaro et al., 1998a; Hanajima et al., 1998; Kujirai et al., 1993; Nakamura et al., 1997) through the activation of an intracortical inhibitory GABAergic circuit (Di Lazzaro et al., 2000; Ilic et al., 2002; Ziemann et al., 1996). In addition, it has been shown that during a slight contraction of the target muscle, SICI is strongly reduced in comparison with the resting condition (Ortu et al., 2008; Rossini et al., 2015).

At the same interstimulus intervals (ISI) a short-interval intracortical facilitation (SICF) occurs if a suprathreshold TS is followed by a subthreshold CS (Tokimura et al., 1996) or, alternatively, when two stimuli near motor threshold are given consecutively (Ziemann, Tergau, Wassermann et al., 1998). The facilitation can be observed at three distinct ISIs after the TS: 1.1–1.5, 2.3–2.9 and 4.1–4.4 ms. It is believed that the first peak of SICF reflects a CS-induced direct excitation of the initial segments of excitatory glutamatergic intracortical interneurons, which had been previously depolarized by the TS-induced EPSPs (Hanajima et al., 2002), while later peaks may represent conventional summation of synaptic inputs at postsynaptic membranes (Hanajima et al., 2002; Ilic et al., 2002). Moreover, the intensities of both TS and CS are able to influence SICF (Ortu et al., 2008; Rossini et al., 2015). Opposite to SICI, a slight active contraction of the target muscle enhances SICF (Ortu et al., 2008).

SICI and SICF are commonly considered as two independent and antagonistic systems (Chen & Garg, 2000; Tokimura et al., 1996; Ziemann, Tergau, Wischer et al., 1998) and their balance may lead to the final result of SICI or SICF protocols (Fisher et al., 2002; Ilic et al., 2002; Ortu et al., 2008; Roshan et al., 2003; Ziemann, Chen et al., 1998). Although SICI and SICF relay on independent neuronal systems, pharmacological studies showed that they are both mediated by $\gamma$-aminobutyric acid type A (GABA$_A$) receptors (Ziemann, 2004), since the first is enhanced and the second is diminished by drugs that increase GABA$_A$ activity (Ziemann, Tergau, Wischer et al., 1998).

A second form of intracortical inhibition can also be studied using TMS. If a suprathreshold TMS pulse is given during voluntary muscle contraction, the motor evoked potential (MEP) is followed by a period of inhibition, known as the cortical silent period (CSP). The duration of the CSP is 100–300 ms and mainly depends on the intensity of TMS rather than the level of muscle contraction (Rossini et al., 2015). Several pharmacological studies have shown that the CSP reflects a long-lasting cortical inhibition mediated through GABA$_B$ receptors (Stetkarova & Kofler, 2013).

SICI, SICF and their interactions have been largely investigated in the limb muscles, mainly in hand muscles,

while in face muscles the former has never been systematically characterized and the latter never explored. Indeed, the few works that investigated SICI in face M1 (Cattaneo & Pavesi, 2014; Paradiso et al., 2005; Pilurzi et al., 2013) used non-comparable ISI and intensities, leading to inconclusive results about this phenomenon in the resting and active face muscles. In view of this, the first aim of the present work was to systematically study SICF and SICI protocols in face M1, testing a large range of ISIs and CS intensities, both at rest and during muscle contraction.

One of the main reasons for studying intracortical circuitry in a face muscle is to compare the results with those in a more frequently studied muscle such as the first dorsal interosseous (FDI). Face muscles are important in a variety of different functions, such as feeding, speech and communication of non-verbal affective states, which require quite different patterns of activity from those of a typical hand muscle. In addition, previous studies have revealed differences in cortical control of face and hand. For example, corticobulbar projections to depressor anguli oris (DAO) are bilateral (Pilurzi et al., 2013) compared with the primarily contralateral projections to FDI. Menon et al. (2018) found that SICI was less strong in bilaterally innervated proximal muscles than in FDI and suggested this might be typical of all muscles with bilateral connectivity. The second aim of the study was therefore to ask whether there are differences in the neurophysiological control of DAO and FDI and if these differences are mediated by GABA$_A$ and/or GABA$_B$ circuits.

Unfortunately, direct comparison of the results of TMS protocols between these muscles is complicated by the fact that their TMS thresholds differ: FDI has a lower threshold than DAO. Given that TMS intensity is known to have dramatic effects on measurements of SICI, SICF and CSP, our third aim was to ask whether such considerations make comparisons impossible (i.e. an 'apples and oranges' problem) or whether some logically valid conclusions can be drawn.

## Methods

### Ethical approval

Experiments were conducted in 15 healthy volunteers (10 females and 5 males; mean age $28.40 \pm 6.31$ (SD) years), all right handed according to the Oldfield Inventory Scale (Oldfield, 1971). All subjects gave their informed written consent to participate in the study, which was approved by the local ethical committee (Bioethics Committee of ASL, no. 1 – Sassari, ID 2075/CE/2014) and conducted in accordance with the *Declaration of Helsinki*. None of the subjects had history or current signs/symptoms of neurological diseases. Subjects sat in a comfortable

chair and were asked to stay relaxed but alert during the experiments.

## Electromyography

EMG was recorded from the right DAO and FDI muscles, using 9 mm-diameter Ag–AgCl surface electrodes. For EMG recordings from the DAO, the active electrode was placed at the midpoint between the angle of the mouth and the lower border of the mandible, the reference electrode over the mandible border, 1 cm below the active electrode, and the ground electrode over the right forehead (Pilurzi et al., 2013, 2020). For EMG recordings from the FDI, the active electrode was placed over the muscle belly, the reference electrode at the second finger metacarpo–phalangeal joint and the ground electrode over the forearm (Rossini et al., 2015). Unrectified EMG signals were recorded (D360 amplifier, Digitimer Ltd, Welwyn Garden City, UK), amplified (×1000), filtered (bandpass 3 to 3000 Hz), sampled (5 kHz per channel; window frame length: 250 ms) using a 1401 power analog-to-digital converter (Cambridge Electronic Design, Cambridge, UK) and Signal 6 software (Cambridge Electronic Design) on a computer and stored for off-line analysis.

## Transcranial magnetic stimulation

TMS was performed using a 70 mm figure-of-eight shaped coil connected to a Magstim 200 stimulator through a Bistim module (Magstim Co., Whitland, Dyfed, UK). The optimal stimulation site, for the right DAO and FDI muscles was carefully searched and then marked with a soft tip pen over the scalp, to maintain the same coil position throughout the experiments. For the DAO the handle of the coil pointed posteriorly and laterally, at approximately 30–45 deg to the interhemispheric line; for FDI the coil pointed backwards and laterally at 45 deg away from the midline (Ginatempo et al., 2021; Ginatempo, Spampinato et al., 2019; Pilurzi et al., 2013, 2020; Rossini et al., 2015). The resting motor threshold (RMT) was taken as the lowest TMS intensity that elicited, in the relaxed muscle, MEPs of 0.05 mV in at least 5 out of 10 consecutive trials and was expressed in percentage of the maximum stimulator output (Rossini et al., 2015). Active motor threshold (AMT) was established as the minimum stimulus intensity able to evoke MEPs >0.2 mV peak-to-peak amplitude in at least 5 out of 10 consecutive trials during isometric contraction of the tested muscle at 10% of maximum voluntary isometric contraction (MVIC) (Rossini et al., 2015). The intensity of the TS for TMS was 120% of RMT or AMT, in resting and active conditions, respectively.

## Experimental design

The physiological proprieties of SICI and SICF were assessed in the DAO and FDI, both in resting and active conditions. The cortical silent period was also compared in both muscles.

**Experiment 1. Short-interval intracortical inhibition of M1 innervating the DAO and FDI muscles at rest.** Rest SICI was studied in all subjects ($n = 15$) from the M1 representation of the right DAO and FDI muscles. Rest SICI was elicited using a paired-pulse TMS protocol with a subthreshold conditioning stimulus (CS) preceding a suprathreshold TS by an ISI of 1.0, 2.0 and 3.0 ms. The CS intensity was set between 50% and 100% of RMT, in steps of 10%, and the TS intensity at 120% of RMT. The experiment was divided up into two blocks: DAO SICI rest and FDI SICI rest. In each subject, the two blocks were randomized and, within each block, all intensities and all states (TS alone and three ISIs) were randomized. Ten unconditioned MEPs and ten conditioned responses for each ISI were recorded. SICI was expressed as the ratio between the conditioned MEP and the unconditioned MEP amplitudes.

**Experiment 2. Short-interval intracortical inhibition of M1 innervating the DAO and FDI muscles during voluntary muscle contraction.** Active SICI was studied in all subjects ($n = 15$) during isometric contraction of the tested muscle at 10% of MVIC from the right DAO and FDI muscles. Active SICI was elicited using a paired-pulse TMS protocol with a subthreshold CS preceding a suprathreshold TS by an ISI of 1.0, 2.0 and 3.0 ms. The CS intensity was set between 50% and 100% of AMT, in steps of 10%, and the TS intensity at 120% of AMT. The experiment was divided up into two blocks: DAO SICI active and FDI SICI active. In each subject, the two blocks were randomized and, within each block, all intensities and all states (TS alone and three ISIs) were randomized. Ten unconditioned MEPs and 10 conditioned responses for each ISI were recorded. SICI was expressed as the ratio of MEP amplitude evoked by the conditioned to the unconditioned MEP.

**Experiment 3. Short-interval intracortical facilitation of M1 innervating the DAO and FDI muscles at rest.** SICF at rest was studied in all subjects ($n = 15$) from the right DAO and FDI muscles. Rest SICF was elicited using a paired-pulse TMS protocol with a sub- and suprathreshold CS succeeding a suprathreshold TS by ISIs of 1.0, 1.5, 2.0, 2.5, 3.0 and 3.5 ms. The CS intensity was set between 80% and 110% of RMT, in steps of 10%, and the TS intensity at 120% of RMT. The experiment was divided up into two blocks: DAO SICF rest and FDI SICF rest. In each subject, the two blocks were randomized and, within

each block, all intensities and all states (TS alone and the five ISIs) were randomized. Ten unconditioned MEPs and 10 conditioned responses for each ISI were recorded. SICF was expressed as the ratio of MEP amplitude evoked by the conditioned to the unconditioned MEP.

**Experiment 4. Short-interval intracortical facilitation of M1 innervating the DAO and FDI muscles during voluntary muscle contraction.** Active SICF was studied in all subjects ($n = 15$) from the M1 representation of the right DAO and FDI muscles during isometric contraction of the tested muscle at 10% of MVIC. Active SICF was elicited using a paired-pulse TMS protocol with a sub- and suprathreshold CS succeeding a suprathreshold TS by ISIs of 1.0, 1.5, 2.0, 2.5, 3.0 and 3.5 ms. The CS intensity was set between 80% and 110% of AMT and the TS intensity at 120% of AMT. The experiment was divided up into two blocks: DAO SICF active and FDI SICF active. In each subject, the two blocks were randomized and, within each block, all intensities and all states (TS alone and the five ISIs) were randomized. Ten unconditioned MEPs and ten conditioned responses for each ISI were recorded. SICF was expressed as the ratio of MEP amplitude evoked by the conditioned to the unconditioned MEP.

**Experiment 5. Cortical silent period of the M1 innervating DAO and FDI muscles.** The cortical silent period (CSP) was investigated in 14 out of 15 subjects from the right DAO and FDI muscles using a single pulse stimulus at an intensity of 120, 130 and 140% AMT. The CSP was recorded in two different conditions: CSP 10% and CSP 100%, during the isometric contraction of the tested muscle at 10% and 100% of MVIC. The experiment was divided up into two blocks: DAO CSP and FDI CSP. The two blocks, condition (10% and 100% of MVIC) and all states (three TS intensities) were randomized in each subject. CSP duration was measured as the time elapsing from the onset of the MEP until the recurrence of voluntary tonic EMG activity.

### Statistical analysis

Statistical analysis was performed with SPSS 20 software (IBM Corp., Armonk, NY, USA). Student's paired $t$-test, repeated measures (RM) analysis of variance (ANOVA) and planned *post hoc* $t$-test with Bonferroni correction for multiple comparison were used. Compound symmetry was evaluated with the Mauchly's test and the Greenhouse–Geisser correction was used when required. Significance was set for $P$-values <0.05. Data are expressed as means ± SD. In all experiments amplitude of conditioned and unconditioned MEPs were analysed. Raw amplitude and ratio were used as variables.

A two-way RM-ANOVA on RMT and AMT intensity values was performed with muscle (DAO and FDI) and condition (rest and active) as within-subjects factors.

For Experiment 1–4, a two-way RM-ANOVA, using raw amplitude as variable, with ISI (Experiment 1 and 2: TS, 1.0, 2.0 and 3.0 ms; Experiment 3 and 4: TS, 1.0, 1.5, 2.0, 2.5, 3.0, and 3.5 ms) and intensity (Experiment 1 and 2: 50%, 60%, 70%, 80%, 90% and 100% RMT or AMT; Experiment 3 and 4: 80%, 90%, 100% and 110% RMT or AMT) as within-subject factors was used separately for DAO and FDI muscles.

To compare the effect between muscles, a three-way RM-ANOVA was performed, separately for each experiment using ratio as variable, with muscle (DAO and FDI), ISI (Experiment 1 and 2: TS, 1.0, 2.0 and 3.0 ms; Experiment 3 and 4: TS, 1.0, 1.5, 2.0, 2.5, 3.0 and 3.5 ms) and intensity (Experiment 1 and 2: 50%, 60%, 70%, 80%, 90% and 100% RMT or AMT; Experiment 3 and 4: 80%, 90%, 100% and 110% RMT or AMT) as within-subject factors.

To compare resting and active conditions, a three-way RM-ANOVA using ratio as variable, with condition (rest and active), ISI (SICI: 1.0, 2.0 and 3.0 ms; SICF: 1.0, 1.5, 2.0, 2.5, 3.0 and 3.5 ms) and intensity (SICI: 50−100% and SICF: 80−110% RMT or AMT, in the resting and active condition, respectively) as within-subject factors was used separately for each muscle and protocol.

For Experiment 5, a three-way RM-ANOVA was performed using muscle (DAO and FDI), MVIC (10% and 100% of the MVIC) and intensity (120%, 130% and 140% of AMT) as within-subject factors.

## Results

Motor thresholds at rest were higher than in the active condition in both muscles. Two-way repeated-measure ANOVA on RMT and AMT intensity values with muscle (DAO and FDI) and condition (rest and active) as within subjects-factor showed higher RMT and AMT for DAO than FDI in both conditions (Table 1). The analysis revealed a significant effect of muscle ($F_{1,14} = 69.419$; $P < 0.001$) and condition ($F_{1,14} = 177.369$; $P < 0.001$) but a non-significant interaction between factors ($F_{1,14} = 1.544$; $P = 0.234$).

### Experiment 1. Short-interval intracortical inhibition of M1 innervating the DAO and FDI muscles at rest

**Resting DAO (Fig. 1A).** In the resting DAO, a significant SICI was observed at subthreshold CS intensities in the 60–80% range of RMT, at all ISIs. In particular, the two-way RM-ANOVA showed a significant effect of ISI ($F_{3,42} = 15.319$; $P < 0.001$), intensity ($F_{5,70} = 7.128$; $P = 0.004$) and interaction between

**Table 1. Neurophysiological parameters of face and hand primary motor cortices**

| Muscles | RMT (%MSO) | AMT (%MSO) | P (RMT vs. AMT) | Rest MEP(mV) | Active MEP(mV) | P (rest MEP vs. active MEP) |
|---|---|---|---|---|---|---|
| DAO | 53.80 ± 6.36 | 44.60 ± 7.08 | <0.001 | 0.20 ± 0.17 | 0.50 ± 0.29 | <0.001 |
| FDI | 39.73 ± 5.90 | 29.53 ± 6.36 | <0.001 | 1.38 ± 0.71 | 1.28 ± 1.00 | 0.40 |
| *P* (DAO vs. FDI) | <0.001 | <0.001 | | <0.001 | <0.001 | |

The table reports means ± standard deviation (SD). MEP amplitude was obtained with 120% RMT at rest and 120% AMT in the active condition. Abbreviations: AMT, active motor threshold; aRMT, resting motor threshold; DAO, depressor anguli oris; FDI, first dorsal interosseus; MEP, motor-evoked potential; MSO, maximum stimulator output.

factors ($F_{15,210} = 5.155$; $P = 0.002$). No significant effect of intensity on MEP amplitude following TS was detected (all $P > 0.05$). A significant inhibition was detected for all the three ISIs compared with TS (1.0 ms: $P = 0.003$; 2.0 ms: $P = 0.007$; 3.0 ms: $P = 0.005$). The *post hoc* analysis of the interaction showed a significant inhibition at 1.0 and 3.0 ms ISI with 60% RMT ($P = 0.001$; $P = 0.028$, respectively), 70% RMT ($P = 0.01$, $P = 0.012$, respectively) and 80% RMT ($P = 0.022$, $P = 0.004$, respectively). At

2.0 ms ISI the inhibition was significant only with a CS of 60% ($P = 0.010$) and 80% RMT ($P = 0.009$).

**Resting FDI (Fig. 1*B*).** In the resting FDI, a significant SICI was observed with all subthreshold and threshold CS intensities, at all ISIs. Two-way RM-ANOVA showed a significant effect of ISI ($F_{342} = 13.911$; $P < 0.001$), intensity ($F_{5,70} = 2.934$; $P = 0.050$) and interaction between factors

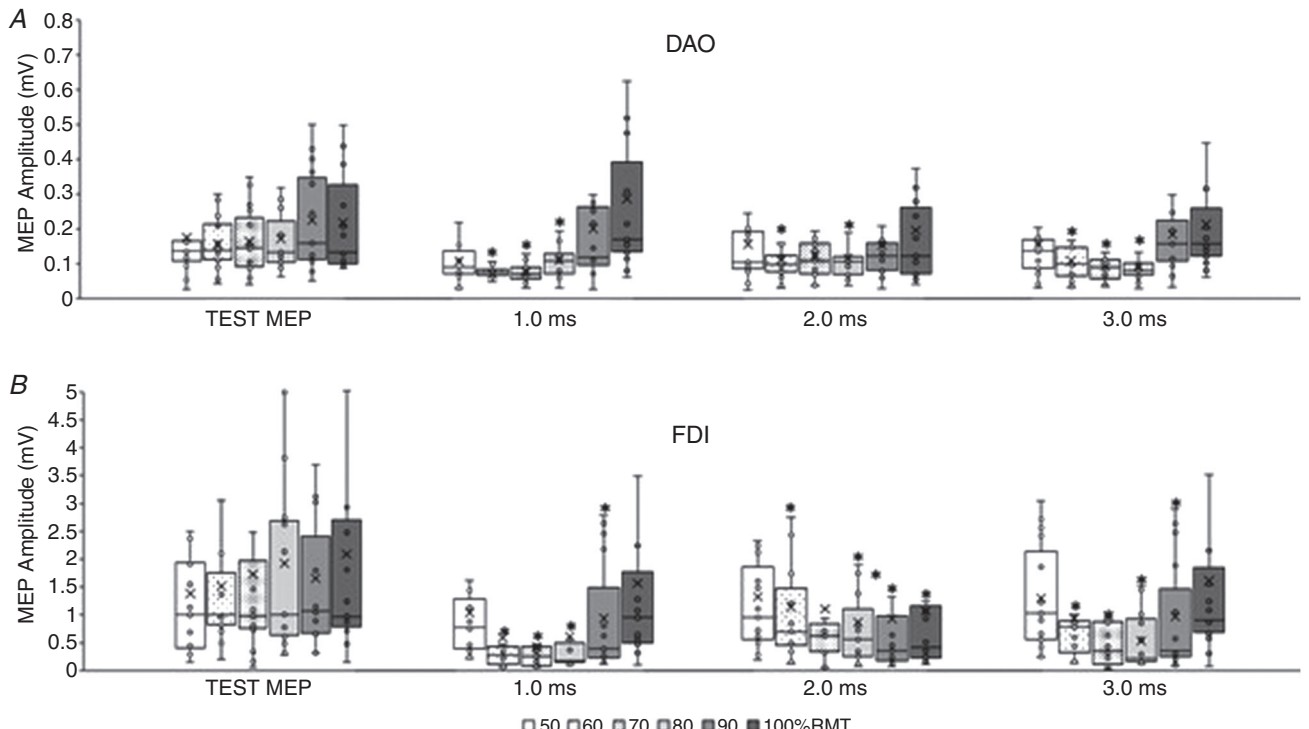

**Figure 1. Short-interval intracortical inhibition (SICI) in face and hand primary motor cortices at rest**
The boxplots report the raw amplitudes of the test MEP, obtained with a single pulse intensity of 120% resting motor threshold (RMT), and of the conditioned MEPs, obtained with intensities of the conditioning stimulus (CS) ranging from 50% to 100% of the RMT and interstimulus intervals (ISI) of 1.0, 2.0 and 3.0 ms. *A*, in the resting depressor anguli oris muscle (DAO), a significant SICI was detected at ISIs of 1.0 and 3.0 ms with CS of 60–80% RMT; at 2.0 ms ISI with a CS of 60% and 80% RMT. *B*, in the resting first dorsalis interosseous muscle (FDI) a clear SICI was observed at all ISIs with CS intensities of 60–90% RMT, while with a CS intensity of 100% RMT a clear SICI was found only at 2.0 ms ISI. The continuous line in the boxplot represents the median value while the '×' symbol represents the mean value of the group. *$P < 0.05$.

($F_{15,210} = 3.933$; $P = 0.010$). No significant effect of intensity on MEP amplitude following TS was detected (all $P > 0.05$). The Bonferroni analysis of the interaction showed a significant inhibition at all ISIs with CS intensities of 60% RMT (1.0 ms: $P = 0.003$; 2.0 ms: $P = 0.008$; 3.0 ms: $P = 0.019$), 70% RMT (1.0 ms: $P = 0.016$; 3.0 ms: $P = 0.027$), 80% RMT (1.0 ms: $P = 0.004$; 2.0 ms: $P = 0.015$; 3.0 ms: $P = 0.005$, respectively) and 90% RMT (1.0 ms; $P = 0.003$; 2.0 ms: $P = 0.006$; 3.0 ms: $P = 0.024$). With the threshold CS intensity (i.e. 100% RMT), a clear SICI was found only at 2.0 ms ISI ($P = 0.015$).

## Experiment 2. Short-interval intracortical inhibition of M1 innervating the DAO and FDI muscles during voluntary muscle contraction

**Active DAO (Fig. 2A).** In the active DAO, a clear SICI was observed for all the ISIs studied but only with a subthreshold CS intensity of 80% AMT. Two-way RM-ANOVA showed a significant effect of ISI ($F_{3,42} = 8.778$; $P = 0.001$), intensity ($F_{5,70} = 10.733$; $P < 0.001$) and interaction between factors ($F_{15,210} = 3.169$; $P = 0.030$). No significant effect of intensity on MEP amplitude following TS was detected (all $P > 0.05$). The *post hoc* analysis of the interaction showed a significant inhibition at 1.0 ms ISI at CS intensities of 50% AMT ($P = 0.022$), 70% AMT ($P = 0.001$) and 80% AMT ($P = 0.016$). At 2.0 and 3.0 ms ISIs a clear SICI was detected only with 80% AMT ($P = 0.009$, $P = 0.041$, respectively).

**Active FDI (Fig. 2B).** In the active FDI, a weak SICI was found. In particular, the two-way RM-ANOVA showed a significant effect of ISI ($F_{3,42} = 4.758$; $P = 0.024$), a non-significant effect of intensity ($F_{5,70} = 2.317$; $P = 0.096$) and no ISI × intensity interaction ($F_{15,210} = 2.118$; $P = 0.117$). No significant effect of intensity on MEP amplitude following TS was detected (all $P > 0.05$). Although the main factor (ISI) showed a significant effect, *post hoc* analysis did not reveal significant differences.

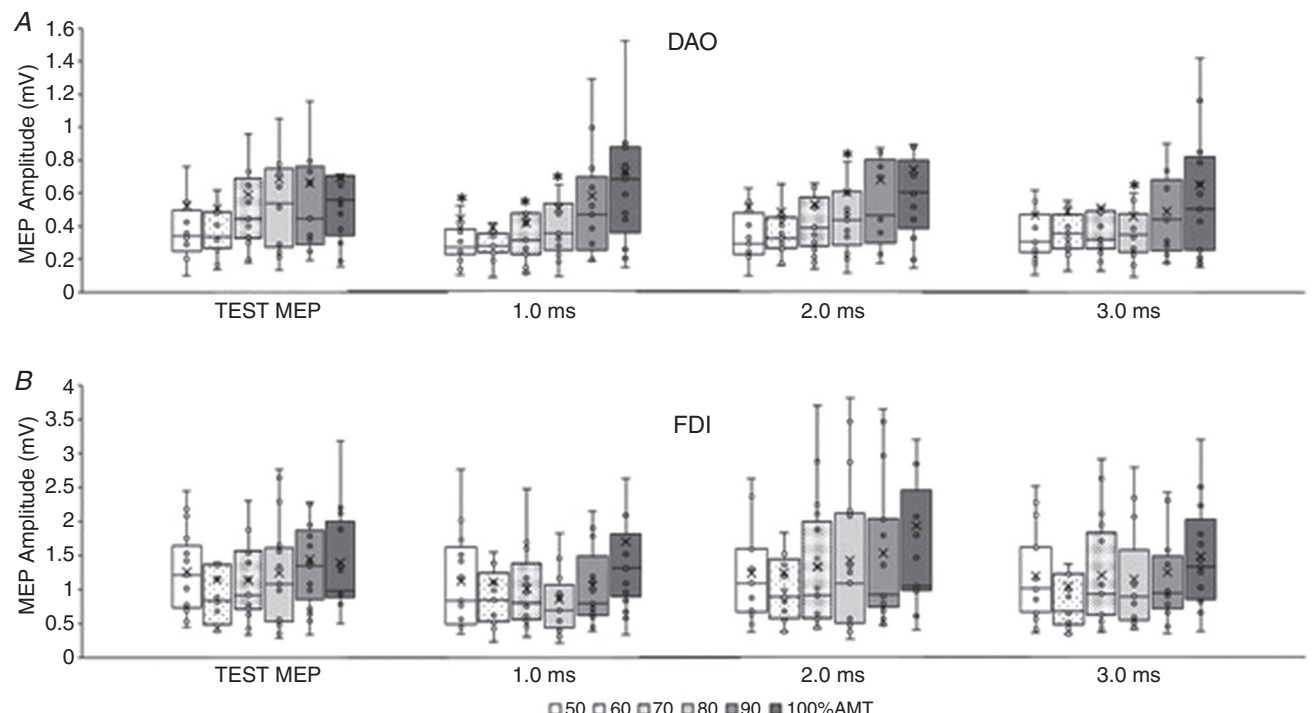

**Figure 2. Short-interval intracortical inhibition (SICI) in face and hand primary motor cortices during voluntary contraction of the target muscles**
The boxplots report the raw amplitudes of the test MEP, obtained with a single pulse intensity of 120% of the active motor threshold (AMT), and of the conditioned MEPs, obtained with intensities of the conditioning stimulus (CS) ranging from 50% to 100% of the AMT, and reported at interstimulus intervals (ISI) of 1.0, 2.0 and 3.0 ms. *A*, in the active depressor anguli oris muscle (DAO), a clear SICI was detected at ISI of 1.0 ms with CS of 50%, 70% and 80% AMT; at 2.0 and 3.0 ms ISIs a clear SICI was detected only with a CS intensity of 80% AMT. *B*, in the active first dorsalis interosseous muscle (FDI) a weak SICI was found only at 1.0 ms ISI. The continuous line in the boxplot represents the median value while the 'x' symbol represents the mean value of the group. *$P < 0.05$.

## Comparison of resting and active of short-interval intracortical inhibition in DAO and FDI muscles

**Resting DAO *vs*. resting FDI (Fig. 3*A* and *C*).** At rest, SICI was stronger in FDI than in DAO. In particular, the three-way RM-ANOVA showed a significant effect of muscle ($F_{1,14} = 5.307$; $P = 0.038$) and intensity ($F_{5,65} = 10.914$; $P < 0.001$) but a non-significant effect of ISI ($F_{2,26} = 2.384$; $P = 0.135$). All interactions were not significant, except for muscle × intensity ($F_{5,65} = 4.060$; $P = 0.009$) and intensity × ISI ($F_{10,130} = 6.063$; $P = 0.001$). The *post hoc* analysis of the intensity × ISI interaction showed a significant difference between 1.0 and 3.0 ms ISIs with CS of 50% RMT ($P = 0.010$) and 60% RMT ($P = 0.025$). A significant difference between 2.0 and 3.0 ms ISIs was detected with CS of 70% RMT ($P = 0.018$) and 80% RMT ($P = 0.017$). FDI showed stronger SICI than DAO at 80% RMT ($P = 0.004$), 90% RMT ($P = 0.015$) and 100% RMT ($P = 0.002$).

**Active DAO *vs*. active FDI (Fig. 3*B* and *D*).** In the active muscle condition, SICI was weaker in FDI than in DAO ($P = 0.034$). The three-way RM-ANOVA showed a significant effect of muscle ($F_{1,14} = 5.591$; $P = 0.034$), intensity ($F_{5,65} = 4.397$; $P = 0.013$) and ISI ($F_{2,26} = 15.409$; $P < 0.001$) but a non-significant interaction among factors. The *post hoc* analysis showed a significant difference between 1.0 and 2.0 ms ISIs ($P = 0.001$) as well as between 2.0 and 3.0 ms ISIs ($P = 0.015$).

**Resting DAO *vs*. active DAO (Fig. 3*A* and *B*).** In the DAO rest SICI was stronger than active SICI. The three-way RM-ANOVA showed a significant effect of condition ($F_{1,14} = 14.452$; $P = 0.002$), ISI ($F_{2,26} = 9.117$; $P = 0.002$) and intensity ($F_{5,65} = 13.113$; $P < 0.001$). Significant interactions were detected as follows: condition × ISI ($F_{5,65} = 12.549$; $P < 0.001$), intensity × ISI ($F_{10,130} = 7.280$; $P < 0.001$) and condition × intensity

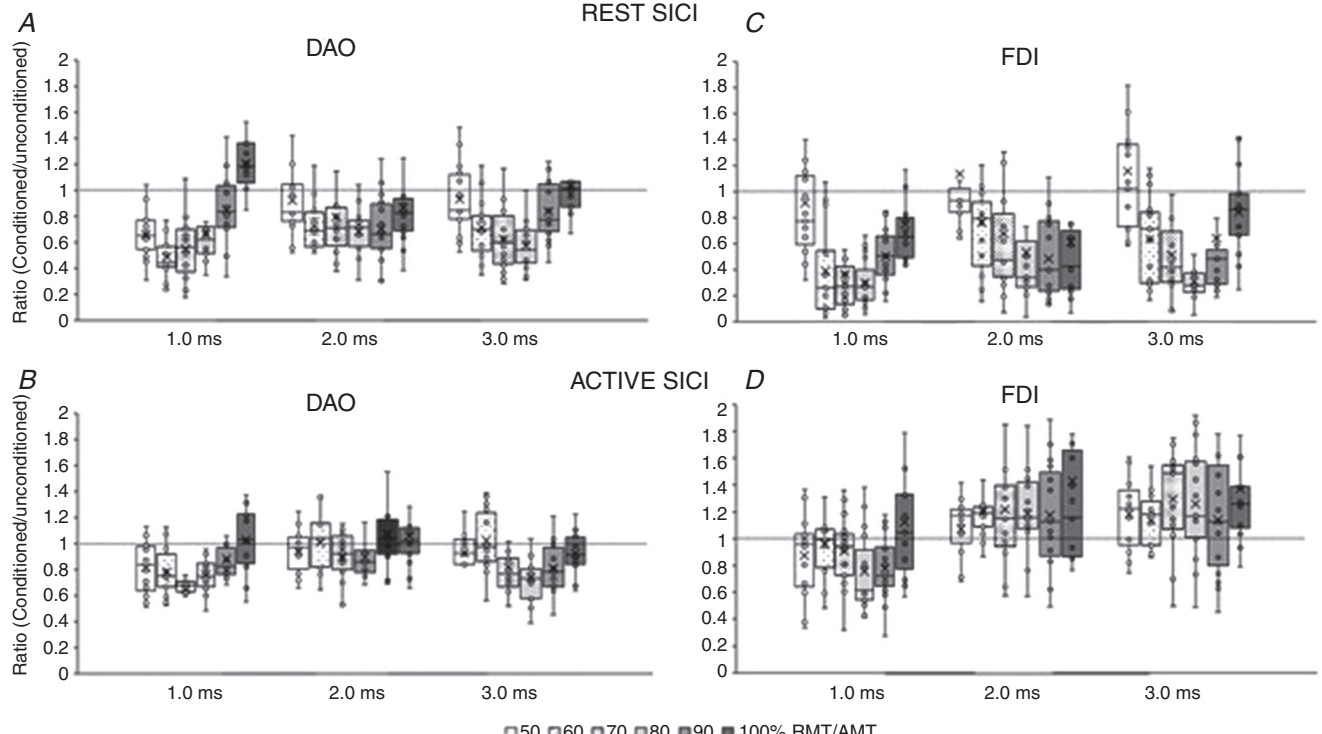

**Figure 3. Comparison of rest and active short-interval intracortical inhibition (SICI) in face and hand primary motor cortices**
The boxplots report conditioned MEP amplitudes expressed as a ratio of the unconditioned MEP (taken as 1.0, horizontal dotted line), induced by the test stimulus alone (120% of the resting motor threshold (RMT) in the resting condition, and 120% of the active motor threshold (AMT) in the active condition). SICI was tested at intensities of the conditioning stimulus (CS) ranging from 50% to 100% of RMT/AMT and is reported at interstimulus intervals (ISI) of 1.0, 2.0 and 3.0 ms. Both the depressor anguli oris muscle (DAO) and first dorsalis interosseous muscle (FDI) showed a stronger SICI at rest (*A* and *C*, respectively) than during the active condition (*B* and *D*, respectively). At rest, DAO (*A*) showed a weaker SICI than the FDI (*C*) at 80–100% RMT. In the active muscles, SICI was stronger in DAO (*B*) than in FDI (*D*). The continuous line in the boxplot represents the median value while the 'x' symbol represents the mean value of the group.

× ISI ($F_{10,130}$ = 3.638; $P$ = 0.006). Bonferroni-adjusted pairwise comparisons showed a significant reduction of SICI at 1 ms ISI at 60% ($P$ = 0.001) and 80% ($P$ = 0.013) RMT/AMT. At 3 ms ISI, SICI was significantly reduced in the active DAO at 60% ($P$ = 0.006), 70% ($P$ = 0.043) and 80% ($P$ = 0.038) RMT/AMT. At 2 ms ISI, the active SICI was significantly reduced at 60% ($P$ = 0.008) and 90% ($P$ = 0.002) RMT/AMT.

**Resting FDI *vs*. active FDI (Fig. 3*C* and *D*).** In the FDI, active SICI was significantly less than rest SICI. The three-way RM-ANOVA showed a significant effect of condition ($F_{1,14}$ = 75.257; $P$ < 0.001), ISI ($F_{2,26}$ = 4.399; $P$ = 0.041) and intensity ($F_{5,65}$ = 7.313; $P$ < 0.001). Significant interactions were found for condition × intensity ($F_{5,65}$ = 5.365; $P$ = 0.003) and condition × intensity × ISI ($F_{10,130}$ = 3.451; $P$ = 0.012). Bonferroni-adjusted pairwise comparisons showed a

significant reduction of active SICI at 1.0 and 2.0 ms ISIs with 60%, 70%, 80%, 90% and 100% RMT/AMT (all $P$ < 0.05). At 3.0 ms ISI, the active SICI was significantly weaker than rest SICI with 60–80% RMT/AMT (all $P$ < 0.05).

## Experiment 3. Short-interval intracortical facilitation of M1 innervating the DAO and FDI muscles at rest

**Resting DAO (Fig. 4*A*).** In the relaxed DAO, a clear SICF was observed at threshold and suprathreshold CS intensities only at the shortest ISIs (1.0 and 1.5 ms). Specifically, the two-way RM-ANOVA showed a significant effect of ISI ($F_{6,84}$ = 10.952; $P$ = 0.001), intensity ($F_{3,42}$ = 5.986; $P$ = 0.007) and a significant interaction between factors ($F_{18,252}$ = 2.306; $P$ = 0.002). No significant effect of intensity on MEP amplitude following TS was detected (all $P$ > 0.05). Bonferroni-adjusted

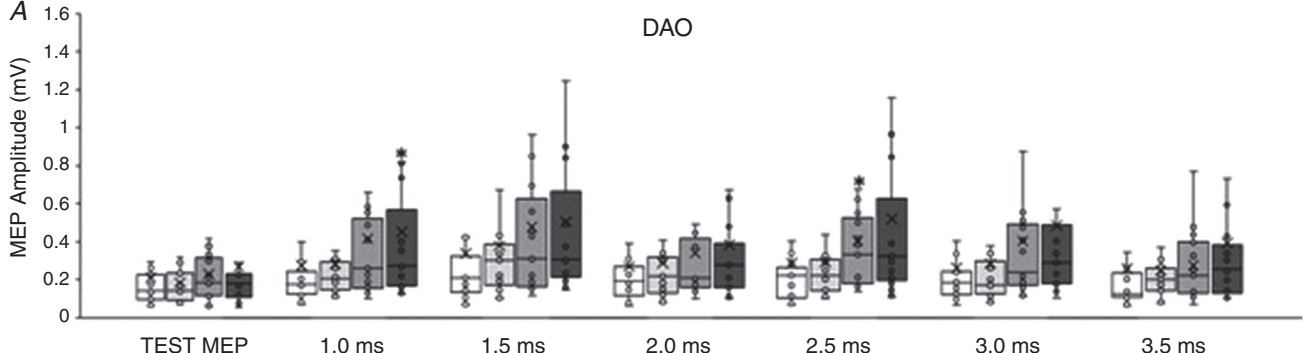

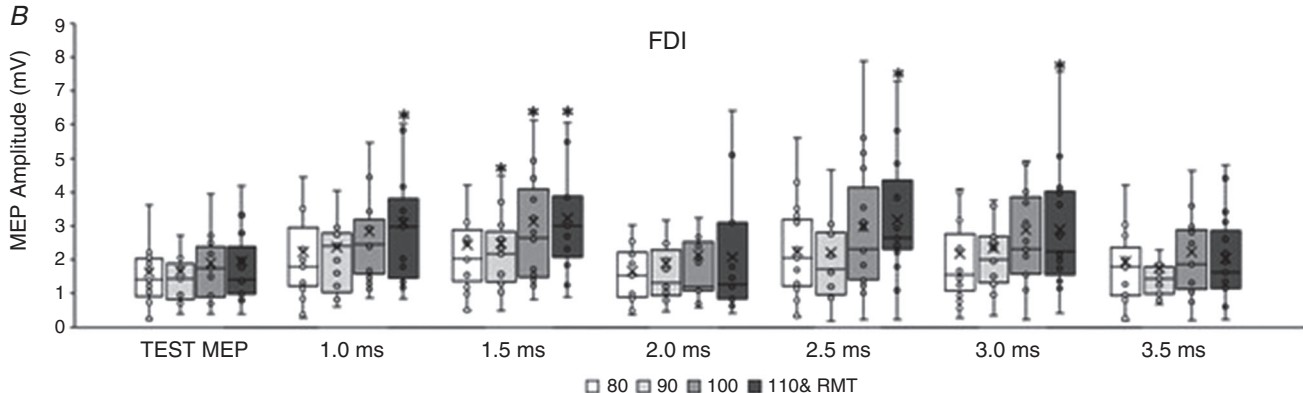

**Figure 4. Short-interval intracortical facilitation (SICF) in face and hand primary motor cortices at rest**
The boxplots report the raw amplitudes of the test MEP obtained with a single pulse intensity of 120% of the resting motor threshold (RMT), and of the conditioned MEPs, obtained with intensities of the conditioning stimulus (CS) ranging from 80% to 110% RMT and interstimulus intervals (ISI) of 1.0, 1.5, 2.0, 2.5, 3.0 and 3.5 ms. *A*, in the resting depressor anguli oris muscle (DAO), a clear SICF was detected with100% RMT at 2.5 ms ISI and with 110% RMT at ISIs of 1.0 and 1.5 ms. *B*, in the resting first dorsalis interosseous muscle (FDI), a clear SICF was observed at ISIs of 1.0 and 1.5 ms with 90–110% RMT while at 3.0 ms ISI, SICF was observed with 100% and 110% RMT. At 2.5 ms ISI, SICF was observed with 100% and 110% RMT. The continuous line in the boxplot represents the median value while the '×' symbol represents the mean value of the group. *$P$ < 0.05.

pairwise comparisons of the interaction revealed a significant facilitation at 100% RMT only at 2.5 ms ISI ($P = 0.029$) and at 110% RMT at 1.0 ms ($P = 0.029$) and a non-significant trend towards facilitation at 1.5 ms ($P = 0.053$) ISIs.

**Resting FDI (Fig. 4*B*).** In the relaxed FDI, a clear SICF was observed with threshold and suprathreshold CS intensities at ISIs of 1.0–1.5 ms and 2.5–3.0 ms. The two-way RM-ANOVA showed a significant effect of ISI ($F_{6,84} = 14.340$; $P < 0.001$), intensity ($F_{3,42} = 4.965$; $P = 0.010$) and interaction between factors ($F_{18,252} = 2.370$; $P = 0.031$). No significant effect of intensity on MEP amplitude following TS was detected (all $P > 0.05$). Bonferroni-adjusted pairwise comparisons of the interactions revealed a significant facilitation with 90% RMT (1.5 ms: $P = 0.029$), 100% RMT (1.5 ms: $P = 0.037$; 2.5 ms: $P = 0.030$; 3.0 ms: $P = 0.018$) and 110% RMT (1.0 ms: $P = 0.031$; 1.5 ms: $P = 0.015$).

## Experiment 4. Short-interval intracortical facilitation of M1 innervating the DAO and FDI muscles during voluntary muscle contraction

**Active DAO (Fig. 5*A*).** A clear SICF was observed at 1.0 and 1.5 ms ISIs at all CS intensities. Two-way RM-ANOVA showed a significant effect of ISI ($F_{6,84} = 12.166$; $P < 0.001$) and intensity ($F_{3,42} = 4.214$; $P = 0.018$) but no significant interaction between factors ($F_{18,252} = 1.646$; $P = 0.162$). No significant effect of intensity on MEP amplitude following TS was detected (all $P > 0.05$). Bonferroni-adjusted *post hoc* tests revealed a significant difference between TS and 1.0 ms ($P = 0.038$), 1.5 ms ($P = 0.032$) and 3.0 ms ($P = 0.027$) ISIs. A stronger facilitation was observed with 110% AMT than 80% AMT ($P = 0.034$).

**Active FDI (Fig. 5*B*).** A clear SICF was detected at 1.5 ms with 100 and 110% AMT. Two-way RM-ANOVA showed

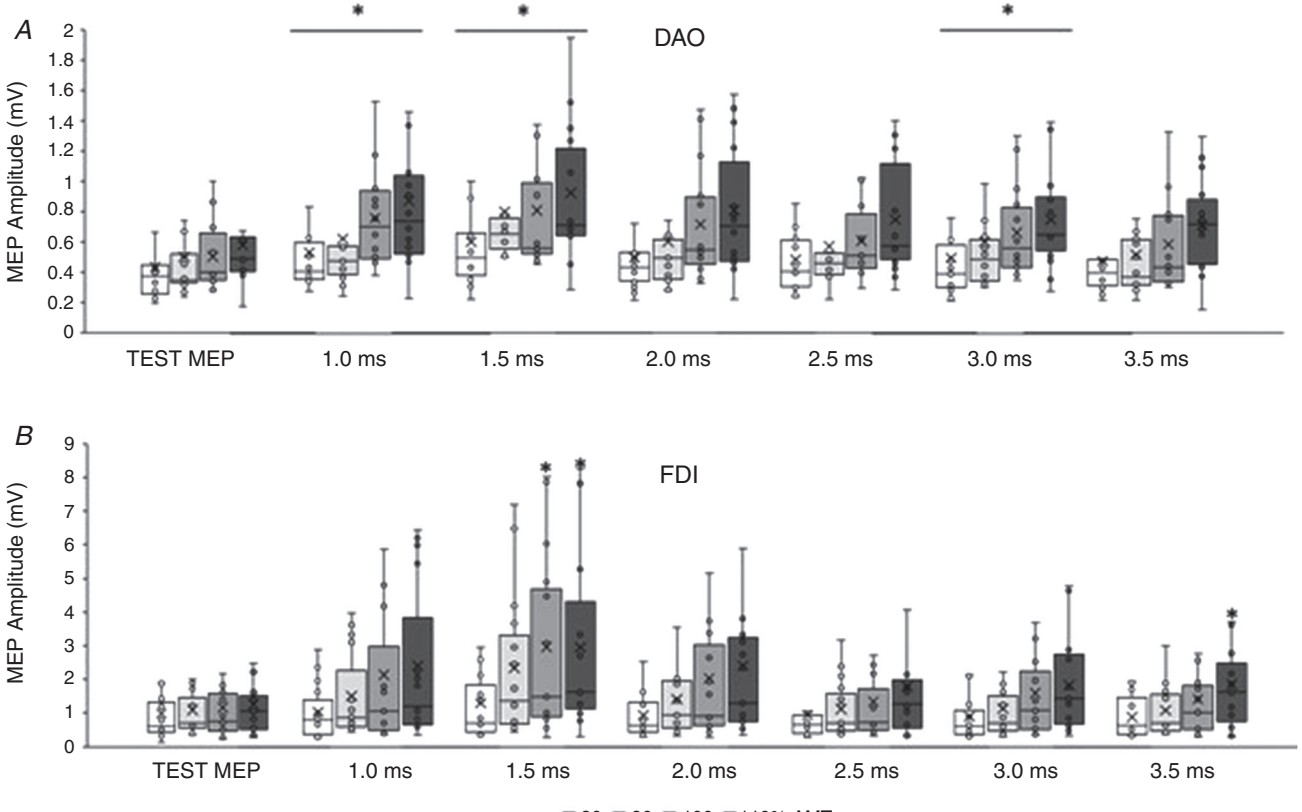

**Figure 5. Short-interval intracortical facilitation (SICF) in face and hand primary motor cortices**
The boxplots report the raw amplitudes of the test MEP, obtained with a single pulse intensity of 120% of the active motor threshold (AMT), and of the conditioned MEPs, obtained with intensities of the conditioning stimulus (CS) ranging from 80% to 110% of the AMT and interstimulus intervals (ISI) of 1.0, 1.5, 2.0, 2.5, 3.0 and 3.5 ms. *A*, in the active depressor anguli oris muscle (DAO), a clear SICF was detected at 1.0, 1.5 and 3.0 ms ISIs. The strongest facilitation was observed with 110% AMT. *B*, in the active first dorsalis interosseous muscle (FDI) a clear SICF was detected at 1.5 ms ISI with 100% and 110% AMT. The continuous line in the boxplot represents the median value while the 'x' symbol represents the mean value of the group. *$P < 0.05$.

a significant effect of ISI ($F_{6,84}$ = 9.565; $P$ = 0.003), intensity ($F_{3,42}$ = 9.977; $P$ = 0.003) and interaction between factors ($F_{18,252}$ = 4.898; $P$ = 0.005). No significant effect of intensity on MEP amplitude following TS was detected (all $P > 0.05$). The *post hoc* analysis showed a significant facilitation at 1.5 ms ISI with 100% AMT ($P$ = 0.048) and 110% AMT ($P$ = 0.035) and at 3.5 ms ISI with 110% AMT ($P$ = 0.041).

### Comparison of resting and active of short-interval intracortical facilitation in DAO and FDI muscles

**Resting DAO *vs*. resting FDI (Fig. 6*A* and *C*).** SICF at rest was similar in FDI and DAO muscles. The three-way RM-ANOVA showed a significant effect of intensity ($F_{3,42}$ = 5.996; $P$ = 0.003) and ISI ($F_{5,70}$ = 21.160; $P < 0.001$) but a non-significant effect of muscle ($F_{1,14}$ = 3.864; $P$ = 0.070) and no interaction among the factors except for intensity $\times$ ISI ($F_{15,210}$ = 2.795;

$P$ = 0.017). The *post hoc* analysis showed a greater facilitation at 100% and 110% RMT for 1.5 and 3.5 ms ISIs (all $P < 0.05$).

**Active DAO *vs*. active FDI (Fig. 6*B* and *D*).** In the active muscle condition, SICF was stronger in FDI than in DAO. Three-way RM-ANOVA revealed a significant effect of intensity ($F_{3,42}$ = 7.145; $P$ = 0.002) and ISI ($F_{5,70}$ = 16.213; $P < 0.001$) but no effect of muscle ($F_{1,14}$ = 2.675; $P$ = 0.126) and no interaction among the factors except for intensity $\times$ ISI ($F_{15,210}$ = 2.885; $P$ = 0.024) and muscle $\times$ ISI ($F_{15,70}$ = 4.996; $P$ = 0.012). Bonferroni-corrected *post hoc* tests revealed that at 1.5 ms ISI the facilitation was stronger in FDI than in DAO ($P$ = 0.012).

**Resting DAO *vs*. active DAO (Fig. 6*A* and *B*).** In the DAO, rest SICF was stronger than active SICF. The three-way RM-ANOVA showed a significant effect of condition ($F_{1,14}$ = 6.627; $P$ = 0.023), ISI ($F_{5,70}$23.321;

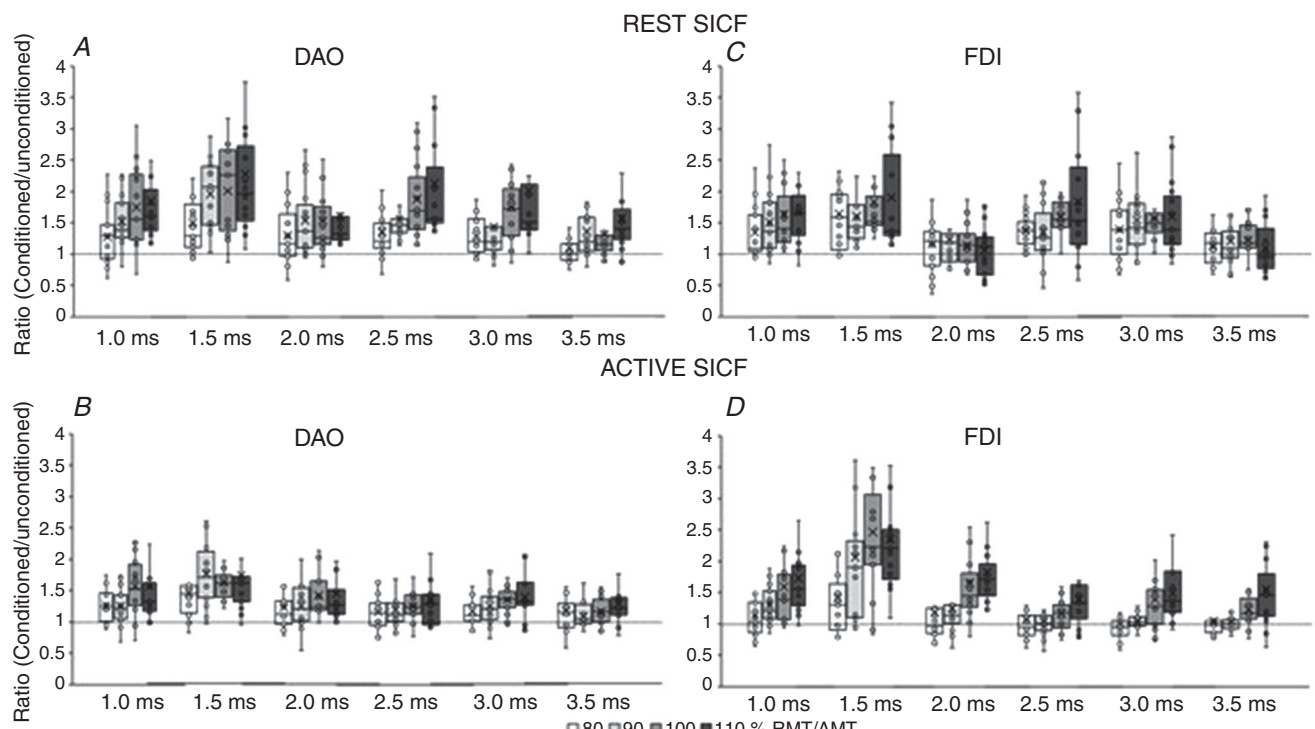

**Figure 6. Comparison of rest and active short-interval intracortical facilitation (SICF) in face and hand primary motor cortices**
The boxplots report conditioned MEP amplitudes expressed as a ratio of the unconditioned MEP (taken as 1.0, horizontal dotted line), induced by the test stimulus alone (120% of the resting motor threshold (RMT) in the resting condition, and 120% of the active motor threshold (AMT) in the active condition). SICF was tested at intensities of the conditioning stimulus (CS) ranging from 80% to 110% RMT/AMT and at interstimulus intervals (ISI) of 1.0, 1.5, 2.0, 2.5, 3.0 and 3.5 ms. In the DAO a stronger facilitation was observed at rest (*A*) than in the active condition (*B*), while no significant difference was observed in FDI between rest (*C*) and active condition (*D*). At rest, SICF was similar in DAO and FDI with 100% and 110% RMT for 1.5 and 3.5 ms ISIs. In the active muscle condition, SICF was significantly different in the two muscles at 1.5 ms ISI. The continuous line in the boxplot represents the median value while the '×' symbol represents the mean value of the group.

$P < 0.001$) and intensity ($F_{3,42} = 4.829$; $P = 0.015$). A significant interaction was observed only for condition × ISI ($F_{15,70} = 3.164$; $P = 0.036$). Bonferroni-adjusted pairwise comparisons showed a stronger facilitation at rest compared with the active state ($P = 0.023$). Moreover, the two conditions were different at 2.5 ms ($P = 0.001$), 3.0 ms ($P = 0.019$) and 3.5 ms ($P = 0.038$) ISIs.

**Resting FDI *vs*. active FDI (Fig. 6*C* and *D*).** In the FDI, no significant difference between rest SICF and active SICF was observed. The three-way RM-ANOVA showed a non-significant effect of condition ($F_{1,14} = 0.004$; $P = 0.953$), and a significant effect of ISI ($F_{5,70} = 23.162$; $P < 0.001$) and intensity ($F_{3,42} = 8.558$; $P < 0.001$). Significant interactions were observed for condition × ISI ($F_{15,70} = 5.617$; $P = 0.007$) and condition × intensity × ISI ($F_{15,210} = 3.369$; $P = 0.004$). Bonferroni-corrected

*post hoc* tests revealed a significant effect of condition at ISIs of 2.0 ms ($P = 0.015$), 2.5 ms ($P = 0.004$) and 3.0 ms ($P = 0.044$). In particular, the significant differences between rest and active conditions (all $P < 0.05$) were observed with 80%, 90% and 100% RMT/AMT at 2.5 ms ISI; with 90% RMT/AMT at 3.0 ms ISI; with 100% and 110% RMT/AMT at 2.0 ms ISI and with 110% RMT/AMT at 3.0 ms ISI.

## Experiment 5. Cortical silent period of the M1 innervating the DAO and FDI muscles

No difference in CSP duration was detected between DAO and FDI (Fig. 7). The three-way RM-ANOVA showed a significant effect of intensity ($F_{2,26} = 40.828$; $P < 0.001$), but a non-significant effect of MVIC ($F_{1,14} = 3.211$; $P = 0.096$) and muscle ($F_{1,14} = 0.156$;

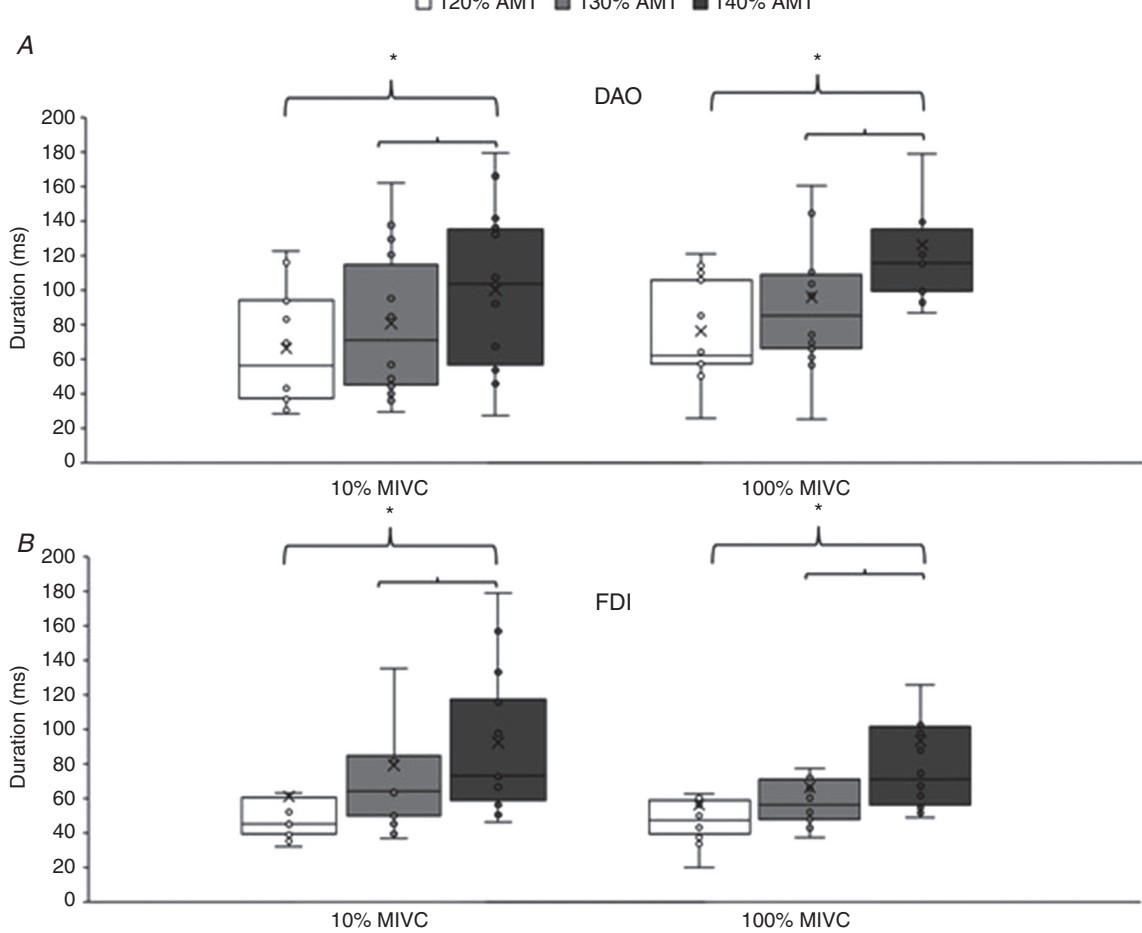

**Figure 7. Cortical silent period (CSP) in face and hand primary motor cortices**
The boxplots report the duration of the CSP in (*A*) the depressor anguli oris muscle (DAO) and (*B*) the first dorsalis interosseous muscle (FDI), tested at stimulus intensity of 120–140% of the active motor threshold (AMT) during activation of the target muscles at 10% (white column) and 100% (black column) of the maximal isometric voluntary contraction (MVIC). In both muscles, the CSP was significantly longer at stimulus intensity of 140% AMT than at 130% AMT and 120% AMT. The continuous line in the boxplot represents the median value while the 'x' symbol represents the mean value of the group. *$P < 0.05$.

$P = 0.699$), and no interaction among the factors. Bonferroni-adjusted pairwise comparisons revealed that the CSP was significantly longer at 140% AMT than 130% AMT ($P < 0.001$) and 120% AMT ($P < 0.001$) in both muscles.

## Discussion

In the present study, SICI and SICF were systematically investigated for the first time in a face muscle (DAO) at rest and during contraction, and the results compared with effects on a hand muscle (FDI) in the same individuals. As expected from previous papers, SICI was weaker in FDI during contraction whereas SICF was stronger. In comparison, resting SICI was smaller in DAO than in FDI and was less affected by voluntary contraction. Resting SICF was similar in both DAO and DFI but was less affected by contraction. We explore some possible reasons for these differences below.

### Rest *vs.* contraction in FDI

A single TMS pulse to motor cortex evokes a series of three or more I-waves, sometimes preceded by a D-wave, in the corticospinal tract (Di Lazzaro, Oliviero et al., 1998). The interval between the I-waves is approximately 1.5 ms, and the number and amplitude of the waves increases with stimulus intensity. The MEP is produced by temporal summation of the EPSPs evoked by the I-waves when they reach spinal motoneurons (Di Lazzaro et al., 2012).

Many previous studies have demonstrated that SICI results from suppression of later I-waves (I3), while early I-waves (I1, I2) are much less affected (Di Lazzaro et al., 2007; Di Lazzaro et al., 2012; Di Lazzaro, Oliviero et al., 1998; Hanajima et al., 1998; Hanajima et al., 2003; Nakamura et al., 1997; Rusu et al., 2014). SICI occurs with conditioning pulse intensities smaller than those needed to recruit I-waves, suggesting that inhibitory neurons have a lower threshold than excitatory neurons. SICF has a higher threshold than SICI and is the result of temporal summation of I-wave volleys produced by the conditioning and test pulses. Thus, SICF is greatest when the I-waves overlap at ISIs around 1.5 and 3 ms (Di Lazzaro et al., 1998a). At these intervals, SICF depends mainly on facilitation of I1 and I2 volleys.

In FDI, SICI is reduced during voluntary contraction. The main reason for this is that voluntary contraction reduces MEP threshold, such that a 1 mV test MEP requires a smaller stimulus than at rest. This recruits fewer late I-waves and SICI appears to be less strong. Basically, MEPs during activation are less dependent on late I-waves than at rest, and it is this that makes SICI appear weaker. A second contribution to reducing SICI occurs at higher conditioning intensities: SICF starts to be recruited and this superimposes on SICI, appearing to reduce its effectiveness. Whether there is also a direct effect on contraction on the SICI itself is unclear.

Similar arguments can be used to understand the effect on contraction on SICF. The MEP is proportionately more dependent on early I-waves and since these are the main contributors to SICF, it appears to be stronger during activation than at rest.

### Comparison of FDI and DAO

At face value, the results show that SICI and SICF behave quite differently in DAO than in FDI. However, simple comparison is complicated by the fact that the (absolute) stimulus intensities used for DAO are higher than FDI. Given that intensity plays such an important role in evaluation of SICI and SICF (Ilic et al., 2002), as illustrated by the effects of voluntary contraction, it is critical to explore the possible consequences in DAO.

The first question is why does the DAO have a higher TMS threshold than FDI?

If we assume that the size and intrinsic properties of the cortical neurons are the same as in the hand motor cortex, and that the lower motoneurons are similarly excitable in each case, there are two main possibilities. The first is because of variations in skull thickness, that face motor cortex could be further from the coil than the hand area. However, if this were the case, then all intensities should scale linearly and when adjusted the behaviour would be the same as in FDI. It would be equivalent to stimulating FDI with a small inert spacer between the coil and scalp surface. A second possible reason for the higher threshold of DAO is that the cortico-bulbar output is less dense than the corticospinal output to FDI (de Noordhout et al., 1999; Palmer & Ashby, 1992). If this were true, it would be necessary to recruit a larger proportion of corticobulbar output to evoke activity in DAO. As a result, we might expect that more late I-waves would be recruited at threshold for DAO than FDI, and in consequence we would expect SICI to be stronger rather than weaker.

But could this be outweighed by the fact that the amplitude of the test MEP in DAO was smaller than in FDI?

For a given intensity of conditioning stimulus, SICI is stronger if the test MEP is large rather than small (Roshan et al., 2003; Sanger et al., 2001). This explanation seems unlikely for two reasons. First, although the DAO test MEP was small, the muscle itself is small and thin. The small MEP may well represent activity in a similar proportion of the motoneuron pool as in FDI. Second, the test stimulus intensity was the same relative to threshold in both muscles and should therefore recruit a similar extra number of late I-waves compared to threshold. Since these are the ones targeted by inhibition, SICI should be similar in both muscles.

Is it possible that reduced SICI in DAO is due to greater overlap with SICF?

This seems unlikely given that we see the differences in SICI at ISI corresponding to the trough of SICF (2 ms). In addition, the threshold difference between SICI and SICF is larger in DAO than in FDI, so that if anything it is less likely that SICI and SICF will overlap.

Considering all these arguments together, it seems highly likely that SICI is less strong at rest in DAO than in FDI.

The intensity of stimulation is also relevant for the question of why contraction has less effect on SICI in DAO than in FDI. Activation reduces the threshold intensity needed to evoke a MEP. However, the threshold reduction is proportionately less in DAO than in FDI (see Table 1). At rest, threshold in DAO may already have recruited late I-waves so that a small reduction in intensity during activation would only reduce their number rather than remove them entirely. The effect on SICI would be less than in FDI, where activation results in a large proportional reduction in stimulus intensity which then would result in a much larger loss of late I-waves and hence reduce SICI.

The DAO–FDI differences in the pattern of SICI–SICF were not observed when looking at the CSP. This discrepancy could be explained by the different neurotransmitter receptors involved in these phenomena, i.e. $GABA_A$ receptors for SICI and SICF (Ziemann, 2004) and $GABA_B$ receptors for CSP (Stetkarova & Kofler, 2013). Consequentially, it is reasonable to interpret the differences observed between face and hand M1 as possibly due to a different control over $GABA_A$ circuits, while the $GABA_B$ system operates in a similar way in the two cortical motor areas.

Finally, why should SICF be weaker in DAO than in FDI?

One possibility is that it is simply due to the fact that SICI, which is activated concurrently during assessment of SICF, is stronger in DAO and tends to cancel SICF. The reason SICF is less affected by voluntary contraction in DAO may be, as above for SICI, that the proportional difference in active and relaxed MEP thresholds is smaller in DAO than in FDI. Thus, SICF changes less in DAO than in FDI during contraction. However, as with SICI, it is not possible to exclude the possibility that in addition to these factors, voluntary contraction also has a direct effect on the SICF itself, increasing its effectiveness in FDI and maintaining or reducing it in DAO.

## Possible reasons for the difference in resting SICI between DAO *vs*. FDI

Although there are a number of reasons (see above) why we should expect SICI to be stronger in DAO

than in FDI, it was, surprisingly, less powerful. One possibility relates to the fact that cortico-bulbar output is bilateral to DAO rather than contralateral as in FDI (Pilurzi et al., 2013). Interestingly, Menon et al. (2018) also noted that SICI was weaker in proximal muscles with a bilateral cortical output compared to FDI. They suggested that SICI could be less powerful in bilateral control, but did not give any reason why this might be advantageous. Face muscles such as DAO are also supposed to be devoid of Renshaw inhibition in the facial motor nucleus (Fanardjian et al., 1983). However, this might have been expected to make cortical inhibitory processes more, rather than less effective; and in any case, Renshaw inhibition in FDI is also very weak or absent compared with that in more proximal muscles (Katz & Pierrot-Deseilligny, 1999).

A second possibility is that SICI itself, i.e. the effectiveness and number of IPSPs produced by the conditioning stimulus, is equally effective in both FDI and DAO. If so, then an intriguing explanation is that recruitment of late I-waves in DAO follows different rules from in FDI. In other words, as the stimulus intensity increases, there is less proportional recruitment of late I-waves in DAO than in FDI. This could also contribute to the higher MEP threshold in DAO. The implication would be that there is a subtle difference in the neural circuitry of the face *versus* the hand area of motor cortex. For example, a reduced tendency to produce reverberating activity might be highly suitable for facial control where rapid, fleeting changes in contraction are highly important methods of non-verbal communication.

## Possible reasons for the difference in active SICI between DAO *vs*. FDI

Active SICI was stronger in DAO than in FDI. This is unlikely to be because of differences in baseline MEP amplitude for all the reasons explained above. During activity the number of late I-waves in active FDI is very much reduced (Di Lazzaro et al. 1998a, 1998b). This is less likely in DAO where the projection is weak (Menon et al., 2018), and late I-waves would be needed to reach threshold even during activity. If so, this might explain why active SICI is stronger in DAO than in FDI: the enhanced late I-wave contribution might over-ride the less effective inhibition that was observed at rest.

Another possibility is that inhibition is facilitated in the active DAO. A possible explanation might reside in the anatomo-physiological differences of face and hand muscle control. In the hand, fine movements are ensured by a reciprocal modulation of inhibitory interneurons at both cortical and spinal level (Fanardjian et al., 1983; Katz & Pierrot-Deseilligny, 1999), with interhemispheric

inhibition playing an important role in the modulation of bilateral hand movements (Ferbert et al., 1992). This does not seem to be the case in facial muscles, where reciprocal inhibition is not as powerful as in the hand (or at least not demonstrated yet) and interhemispheric inhibition is absent (Ginatempo et al., 2021; Ginatempo, Manzo et al., 2019). The result may be that enhanced activity in intracortical inhibitory interneurons of face M1 has to compensate for these other forms of inhibition to ensure fine control of the complex facial muscle system. Such a mechanism would control the spread of excitation and allow fractionation of muscle activity. It might also enhance rapid termination of contraction. Both features are required to produce rapid and subtle changes in facial expression that are characteristic of both verbal and non-verbal communication (Müri, 2016).

## Conclusion

How far is it possible to compare responses to TMS protocols between muscles when the muscles themselves have different thresholds? Are they apples and oranges that should never be compared, or is it possible to improve on that? In this paper we have explored some of the possible confounding factors and conclude that these may well account for some of the differences between DAO and FDI (e.g. effect of contraction on the efficiency of SICI and SICF), but also argue that substantial 'real' differences may exist (e.g. less strong resting SICI in DAO).

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

## Additional information

### Data availability statement

The data that support the findings of this study are available from the corresponding author upon request.

### Competing interests

The authors do not have any competing interest in and did not receive any funding for this research.

### Author contributions

The experiments were performed at the laboratories of neurophysiology of the Department of Biomedical Sciences, University of Sassari, Sassari (Italy). Conception and design of the experiments: F.G. and F.D.; acquisition, analysis and interpretation of data: F.G., N.L., A.M., J.C.R. and F.D.; drafting the article or revising it critically for important intellectual content: F.G., N.L., A.M., J.C.R. and F.D. All authors approved the final version for publication, agree to be accountable for all

aspects of the work in ensuring that questions related to the accuracy or integrity of any part of the work are appropriately investigated and resolved. All persons designated as authors qualify for authorship, and all those who qualify for authorship are listed.

## Acknowledgements

Open Access Funding provided by Universita degli Studi di Sassari within the CRUI-CARE Agreement.

## Funding

This work was supported by the Banco di Sardegna Foundation 2017 call ('Bando competitivo Fondazione di Sardegna – 2017) and by the Fondo di Ateneo per la Ricerca (FAR) 2019 and 2020. N.L. was supported by a PhD grant in Biomedical Sciences.

## Keywords

face muscles, face primary motor cortex, hand primary motor cortex, intracortical facilitation, intracortical inhibition, TMS

## Supporting information

Additional supporting information can be found online in the Supporting Information section at the end of the HTML view of the article. Supporting information files available:

**Peer Review History**
**Statistical Summary Document**

