## [Peer Review History · The Journal of Physiology]

Is it possible to compare inhibitory and excitatory intracortical circuits in face and hand primary motor cortex?

Francesca Ginatempo, Nicola Loi, Andrea Manca, John C Rothwell, and Franca Deriu

DOI: 10.1113/JP283137

Corresponding author(s): Franca Deriu (deriuf@uniss.it)

The following individual(s) involved in review of this submission have agreed to reveal their identity: Jason Neva (Referee #1)

Review Timeline:

Submission Date:	22-Sep-2021
Editorial Decision:	05-Jan-2022
Resubmission Received:	23-Mar-2022
Editorial Decision:	12-May-2022
Revision Received:	31-May-2022
Accepted:	13-Jun-2022

Senior Editor: Richard Carson

Reviewing Editor: Dario Farina

Transaction Report:

Dear Professor Deriu,

Re: JP-RP-2021-282295 "Inhibitory and excitatory intracortical circuits function differently in face and hand primary motor cortex." by Francesca Ginatempo, Nicola Loi, John C Rothwell, and Franca Deriu

Thank you for submitting your manuscript to The Journal of Physiology. It has been assessed by a Reviewing Editor and by 2 Referees and the reports are copied below.

Please let your co-authors know of the following editorial decision as quickly as possible.

As you will see, in its current form, the manuscript is not acceptable for publication in The Journal of Physiology. In comments to me, the Reviewing Editor expressed interest in the potential of this study, but much work still needs to be done (and this may include new experiments) in order to satisfactorily address the concerns raised in the reports.

In view of this interest, I would like to offer you the opportunity to carry out all of the changes requested in full, and to resubmit a new manuscript using the "Submit Special Case Resubmission for JP-RP-2021-282295..." on your homepage.

We cannot, of course, guarantee ultimate acceptance at this stage as the revisions required are substantial. However, we encourage you to consider the requested changes and resubmit your work to us if you are able to complete or address all changes.

A new manuscript would be renumbered and redated, but the original referees would be consulted wherever possible. An additional referee's opinion could be sought, if the Reviewing Editor felt it necessary. A full response to each of the reports should be uploaded with a new version.

I hope that the points raised in the reports will be helpful to you.

Yours sincerely,

Richard Carson
Senior Editor
The Journal of Physiology

EDITOR COMMENTS

Reviewing Editor:

The two reviewers had slightly different opinions on the overall quality and impact of the study. Nonetheless, both highlighted concerns on the current presentation of the work. The main issues identified by the reviewers are the weak motivation for the study (including for the choice of muscles) and lack of a theoretical basis that could support a solid hypothesis. Another major concern is related to possible differences between baselines in the conditions compared, which need to be clarified (perhaps with additional measures). Both reviewers also noted the large number of multiple comparisons not statistically corrected. Overall, while the study is well conducted and of potential interest for The Journal of Physiology, a substantial revision, which may include new measures and analyses, is needed.

Senior Editor:

Please ensure that there is compliance with instructions provided in the Information for Authors, prior to submission.

REFEREE COMMENTS

Referee #1:

Reviewer Response to Manuscript JP-RP-2021-282295

The authors present an interesting and necessary study investigating the intracortical mechanisms underlying the face (DOA) and hand muscles (FDI). Specifically, they measured short-interval intracortical inhibition (SICI) and short-interval intracortical facilitation (SICF). They found that SICI was stronger in FDI at rest compared to DOA, but the opposite with muscle contraction. They also found that SICF was present to a similar extent at rest for both muscles (DOA and FDI), but with muscle contraction it was stronger in FDI compared to DOA. This is important work to consider the fundamental

inhibitory and facilitatory circuits underpinning corticospinal excitability of the face and hand muscles, and their interactions. This is a well-done study, with a strong rationale, relatively clear results and I believe is appropriate for the readership of the Journal of Physiology. I have a few comments regarding the presentation of results and statistical analysis, which I outline below. The authors do an excellent job of discussing the fundamental implications of the results in the Discussion.

Major

1. The results and figures are quite complicated, and they could benefit from drawing lines to show the most important differences between ISIs and CS intensities to guide the reader.
2. Further to the point above, if the authors could find a way to simplify the presentation of the results (i.e., highlight the most important findings, according to the authors' interpretation of their results), this would greatly increase the readership and understanding of these important findings.
3. At times in the results a significant or a weak effect was reported when the $p > 0.05$ (for example, $p = 0.053$ for an effect of SICF). I recommend that these be reported as non-significant findings, particularly considering all of the CS intensities, ISIs, etc.
4. Further to the point above, it may be appropriate to consider correction for multiple comparisons of the different conditions (active DOA, rest DOA, active FDI, rest FDI).

Minor

1. Certain phrases throughout the manuscript could be corrected for English grammar. Some examples include:
 - a. Abstract: "SICF had similar extent..." should read like "SICF increased/decreased to a similar extent...." Or it could read like "SICF had a similar increase/decrease in the DOA and FDI..."
 - b. Introduction: "Therefore, in the present work we investigated systematically SICI and SICF..." would read much better if written like "Therefore, in the present work we systematically investigated SICI and SICF..."
 - c. Results: "In the FDI, active SICI was significantly lesser than rest SICI." should read "In the FDI, active SICI was significantly less than rest SICI" or "significantly reduced compared to rest SICI"
 - d. Titles of the results sections say "the M1" where it would be better to simply write M1 without "the".
 - e. Results: "At rest, DAO and FDI showed similar SICF effect." Should read "At rest, DAO and FDI showed similar SICF."

Referee #2:

In this study, the authors investigate the inhibitory and excitatory properties of M1 face and hand intracortical circuits. Specifically, they use paired-pulse TMS protocols to examine short-interval intracortical inhibition (SICI) and short-interval intracortical facilitation (SICF) of the M1 representation of face muscle, depressor anguli (DAO), and hand muscle, first dorsal interosseous (FDI). The results seem to suggest that SICI was stronger in FDI than DAO at rest, but the opposite is true during muscle contraction; while for SICF, FDI showed higher modulation than DAO during muscle contraction. The study appears to be well designed and carefully conducted. I have the following comments on the study. Hopefully, they can help to improve the quality of the paper.

My main concern is if the effect was mainly driven by baseline MEP, rather than the inhibitory/excitatory property differences in the face and hand areas. Resting and active MEP amplitudes without conditioning as well as RMT and AMT values for the two muscles need to be reported. As the authors mentioned in the discussion that the test MEPs for FDI were $>1.0\text{mV}$, and those for DAO were $<0.5\text{mV}$. This is true for both SICI and SICF. Directly comparing these two muscles seem to be like comparing apples and oranges.

Many of the interpretations seem to be post hoc. What were the authors' predictions, and what are the theoretical bases for those predictions? It's fine if the authors make it clear that this is an exploratory study. While it's interesting to see the difference between face and hand inhibitory and excitatory circuitries, the motivation of directly comparing face and hand muscles is not clear. For example, why not compare face and leg or hand and leg? Given that the two muscles are functionally so different in many ways, what would be the surprise that they show different inhibitory and excitatory properties?

It is not clear to me why would SICI have an overlap with SICF in FDI, but not in DAO. This again seems to be a post hoc interpretation.

The fourth paragraph of the Introduction does not seem to be relevant to the current topic. And CSP was not mentioned at all. Experiment 5 came out of the blue.

There are so many statistical tests that it may warrant a correction for multiple comparisons.

The resolution of the figures is too low, making it difficult to read.

ADDITIONAL FORMATTING REQUIREMENTS:

-Author photo and profile. First (or joint first) authors are asked to provide a short biography (no more than 100 words for one author or 150 words in total for joint first authors) and a portrait photograph. These should be uploaded and clearly labelled with the revised version of the manuscript. See Information for Authors for further details.

-Your manuscript must include a complete Additional Information section

-A Statistical Summary Document, summarising the statistics presented in the manuscript, is required upon revision. It must be on the Journal's template, which can be downloaded from the link in the Statistical Summary Document section here: https://jp.msubmit.net/cgi-bin/main.plex?form_type=display_requirements#statistics

-Papers must comply with the Statistics Policy https://jp.msubmit.net/cgi-bin/main.plex?form_type=display_requirements#statistics

In summary:

-If $n \leq 30$, all data points must be plotted in the figure in a way that reveals their range and distribution. A bar graph with data points overlaid, a box and whisker plot or a violin plot (preferably with data points included) are acceptable formats.

-If $n > 30$, then the entire raw dataset must be made available either as supporting information, or hosted on a not-for-profit repository e.g. FigShare, with access details provided in the manuscript.

- n clearly defined (e.g. x cells from y slices in z animals) in the Methods. Authors should be mindful of pseudoreplication.

-All relevant n values must be clearly stated in the main text, figures and tables, and the Statistical Summary Document (required upon revision)

-The most appropriate summary statistic (e.g. mean or median and standard deviation) must be used. Standard Error of the Mean (SEM) alone is not permitted.

-Exact p values must be stated. Authors must not use 'greater than' or 'less than'. Exact p values must be stated to three significant figures even when 'no statistical significance' is claimed.

-Statistics Summary Document completed appropriately upon revision

-A Data Availability Statement is required for all papers reporting original data. This must be in the Additional Information section of the manuscript itself. It must have the paragraph heading "Data Availability Statement". All data supporting the results in the paper must be either: in the paper itself; uploaded as Supporting Information for Online Publication; or archived in an appropriate public repository. The statement needs to describe the availability or the absence of shared data. Authors must include in their Statement: a link to the repository they have used, or a statement that it is available as Supporting

Information; reference the data in the appropriate sections(s) of their manuscript; and cite the data they have shared in the References section. Whenever possible the scripts and other artefacts used to generate the analyses presented in the paper should also be publicly archived. If sharing data compromises ethical standards or legal requirements then authors are not expected to share it, but must note this in their Statement. For more information, see our Statistics Policy.

Confidential Review

22-Sep-2021

Reviewer Response to Manuscript JP-RP-2021-282295

The authors present an interesting and necessary study investigating the intracortical mechanisms underlying the face (DOA) and hand muscles (FDI). Specifically, they measured short-interval intracortical inhibition (SICI) and short-interval intracortical facilitation (SICF). They found that SICI was stronger in FDI at rest compared to DOA, but the opposite with muscle contraction. They also found that SICF was present to a similar extent at rest for both muscles (DOA and FDI), but with muscle contraction it was stronger in FDI compared to DOA. This is important work to consider the fundamental inhibitory and facilitatory circuits underpinning corticospinal excitability of the face and hand muscles, and their interactions. This is a well-done study, with a strong rationale, relatively clear results and I believe is appropriate for the readership of the Journal of Physiology. I have a few comments regarding the presentation of results and statistical analysis, which I outline below. The authors do an excellent job of discussing the fundamental implications of the results in the Discussion.

Major

1. The results and figures are quite complicated, and they could benefit from drawing lines to show the most important differences between ISIs and CS intensities to guide the reader.
2. Further to the point above, if the authors could find a way to simplify the presentation of the results (i.e., highlight the most important findings, according to the authors interpretation of their results), this would greatly increase the readership and understanding of these important findings.
3. At times in the results a significant or a weak effect was reported when the $p > 0.05$ (for example, $p = 0.053$ for an effect of SICF). I recommend that these be reported as non-significant findings, particularly considering all of the CS intensities, ISIs, etc.
4. Further to the point above, it may be appropriate to consider correction for multiple comparisons of the different conditions (active DOA, rest DOA, active FDI, rest FDI).

Minor

1. Certain phrases throughout the manuscript could be corrected for English grammar. Some examples include:
 - a. Abstract: "SICF had similar extent..." should read like "SICF increased/decreased to a similar extent...." Or it could read like "SICF had a similar increase/decrease in the DOA and FDI..."
 - b. Introduction: "Therefore, in the present work we investigated systematically SICI and SICF..." would read much better if written like "Therefore, in the present work we systematically investigated SICI and SICF..."
 - c. Results: "In the FDI, active SICI was significantly lesser than rest SICI." should read "In the FDI, active SICI was significantly less than rest SICI" or "significantly reduced compared to rest SICI"
 - d. Titles of the results sections say "the M1" where it would be better to simply write M1 without "the".
 - e. Results: "At rest, DAO and FDI showed similar SICF effect." Should read "At rest, DAO and FDI showed similar SICF."

Editor and Reviewer Response to Manuscript JP-RP-2021-282295

EDITOR COMMENTS

Reviewing Editor:

The two reviewers had slightly different opinions on the overall quality and impact of the study. Nonetheless, both highlighted concerns on the current presentation of the work. The main issues identified by the reviewers are the weak motivation for the study (including for the choice of muscles) and lack of a theoretical basis that could support a solid hypothesis. Another major concern is related to possible differences between baselines in the conditions compared, which need to be clarified (perhaps with additional measures). Both reviewers also noted the large number of multiple comparisons not statistically corrected. Overall, while the study is well conducted and of potential interest for The Journal of Physiology, a substantial revision, which may include new measures and analyses, is needed.

We thank the Editor for summarizing the Reviewers' comments and giving us the opportunity to revise the manuscript according to their suggestions. As a result, we have included an in-depth consideration of the problems of comparing such TMS measures across muscles with such different baseline thresholds and CMAP amplitudes. The paper now has an amended title, and new abstract, with major modifications to the Introduction and Discussion.

Senior Editor:

Please ensure that there is compliance with instructions provided in the Information for Authors, prior to submission.

Done. Thank you.

REFEREE COMMENTS

Referee #1:

The authors present an interesting and necessary study investigating the intracortical mechanisms underlying the face (DAO) and hand muscles (FDI). Specifically, they measured short-interval intracortical inhibition (SICI) and short-interval intracortical facilitation (SICF). They found that SICI was stronger in FDI at rest compared to DAO, but the opposite with muscle contraction. They also found that SICF was present to a similar extent at rest for both muscles (DAO and FDI), but with muscle contraction it was stronger in FDI compared to DAO. This is important work to consider the fundamental inhibitory and facilitatory circuits underpinning corticospinal excitability of the face and hand muscles, and their interactions. This is a well-done study, with a strong rationale, relatively clear results and I believe is appropriate for the readership of the Journal of Physiology. I have a few comments regarding the presentation of results and statistical analysis, which I outline below. The authors do an excellent job of discussing the fundamental implications of the results in the Discussion.

Major

1. The results and figures are quite complicated, and they could benefit from drawing lines to show the most important differences between ISIs and CS intensities to guide the reader.

We agree that line graphs are more readable and, therefore, we have changed the figures according to the Reviewer's suggestion.

2. Further to the point above, if the authors could find a way to simplify the presentation of the results (i.e., highlight the most important findings, according to the authors interpretation of their results), this would greatly increase the readership and understanding of these important findings.

We apologize for being confusing in reporting the results. Following the Reviewer's suggestion, each section of the results is now introduced by a simple sentence summarizing the main findings.

3. At times in the results a significant or a weak effect was reported when the $p > 0.05$ (for example, $p = 0.053$ for an effect of SICF). I recommend that these be reported as non-significant findings, particularly considering all of the CS intensities, ISIs, etc.

Done.

4. Further to the point above, it may be appropriate to consider correction for multiple comparisons of the different conditions (active DAO, rest DAO, active FDI, rest FDI).

We apologize for being unclear on this important issue. The Bonferroni correction was used for the multiple comparisons but we incorrectly reported the degrees of freedom. The correct values are now reported along with the F values of the ANOVAs in the results.

Minor

1. Certain phrases throughout the manuscript could be corrected for English grammar. Some examples include:

The Manuscript has been revised by a native English person.

a. Abstract: "SICF had similar extent..." should read like "SICF increased/decreased to a similar extent..." Or it could read like "SICF had a similar increase/decrease in the DOA and FDI..."

Done (lines: 54-55).

b. Introduction: "Therefore, in the present work we investigated systematically SICI and SICF..." would read much better if written like "Therefore, in the present work we systematically investigated SICI and SICF..."

Done.

c. Results: "In the FDI, active SICI was significantly lesser than rest SICI." should read "In the FDI, active SICI was significantly less than rest SICI" or "significantly reduced compared to rest SICI"

Done (lines: 304-305).

d. Titles of the results sections say "the M1" where it would be better to simply write M1 without "the".

Done.

e. Results: "At rest, DAO and FDI showed similar SICF effect." Should read "At rest, DAO and FDI showed similar SICF."

Done (line: 347-348).

Referee #2:

In this study, the authors investigate the inhibitory and excitatory properties of M1 face and hand intracortical circuits. Specifically, they use paired-pulse TMS protocols to examine short-interval intracortical inhibition (SICI) and short-interval intracortical facilitation (SICF) of the M1 representation of face muscle, depressor anguli (DAO), and hand muscle, first dorsal interosseous (FDI). The results seem to suggest that SICI was stronger in FDI than DAO at rest, but the opposite is true during muscle contraction; while for SICF, FDI showed higher modulation than DAO during muscle contraction. The study appears to be well designed and carefully conducted. I have the following comments on the study. Hopefully, they can help to improve the quality of the paper. My main concern is if the effect was mainly driven by baseline MEP, rather than the inhibitory/excitatory property differences in the face and hand areas. Resting and active MEP amplitudes without conditioning as well as RMT and AMT values for the two muscles need to be reported.

We apologize for not reporting this information, which has been now added in a new Table 1 (line 238; Table 1).

As the authors mentioned in the discussion that the test MEPs for FDI were $>1.0\text{mV}$, and those for DAO were $<0.5\text{mV}$. This is true for both SICI and SICF. Directly comparing these two muscles seem to be like comparing apples and oranges.

This is indeed a very important consideration and following the Reviewer's suggestion we have taken the opportunity to address it in far more detail in the amended text. Accordingly, a large number of changes have been made including the Title, Abstract, Introduction and Discussion. We now consider not only the differences in MEP amplitude, but also differences in threshold and possible I-wave composition, all of which can affect the interpretation of the TMS measures we have used here.

Many of the interpretations seem to be post hoc. What were the authors' predictions, and what are the theoretical bases for those predictions? It's fine if the authors make it clear that this is an exploratory study. While it's interesting to see the difference between face and hand inhibitory and excitatory circuitries, the motivation of directly comparing face and hand muscles is not clear. For example, why not compare face and leg or hand and leg? Given that the two muscles are functionally so different in many ways, what would be the surprise that they show different inhibitory and excitatory properties?

We thank the Reviewer for allowing us to clarify this important point.

Our study was in fact motivated by exactly the same considerations as the reviewer has outlined. These are two muscles which have very different functions, and the question is whether their cortical control will reflect that difference. Although we suspected there should be some differences, we would argue that this is not necessarily the case: the basic neurophysiology could have been similar, even if the control signals differ. Our results suggest that some differences may indeed be present.

We acknowledge that we could have included a lower limb muscle in addition to hand and face. However, there is far more existing data on leg muscle neurophysiology than for the face.

It is not clear to me why would SICI have an overlap with SICF in FDI, but not in DAO. This again seems to be a post hoc interpretation.

Since no data are available in the literature about the presence and the features of SICF in face M1, we could only relay on the well-known overlap between SICF and SICI described in the FDI (Peurala et al., 2008; Ortu et al. 2008), which is considered responsible for the absence of SICI in active FDI. Our results showed that in the DAO this overlap is unlikely to happen, because both SICI and SICF were clearly detected at separate intensities and ISIs. This point has been better discussed in the revised Manuscript (Lines 447-450).

The fourth paragraph of the Introduction does not seem to be relevant to the current topic. And CSP was not mentioned at all. Experiment 5 came out of the blue.

According to the Reviewer's suggestion, the fourth paragraph has been deleted. Regarding the CSP, we apologize for not mentioning it in the Introduction. In the revised manuscript the rationale for testing the CSP has been added (Lines 85-91).

There are so many statistical tests that it may warrant a correction for multiple comparisons.

We apologize for being unclear on this important issue. The Bonferroni correction was used for the multiple comparisons but we incorrectly reported the degrees of freedom. Their correct values are now correctly reported along the F values in the results.

The resolution of the figures is too low, making it difficult to read.

The resolution of the figures has been improved according to the Reviewer's comment and they have been changed from histograms to line graphs to address one of the suggestions of Reviewer #1.

Dear Professor Deriu,

Re: JP-RP-2022-283137X "Is it possible to compare inhibitory and excitatory intracortical circuits in face and hand primary motor cortex?" by Francesca Ginatempo, Nicola Loi, Andrea Manca, John C Rothwell, and Franca Deriu

Thank you for resubmitting your manuscript to The Journal of Physiology. It has been assessed by a Reviewing Editor and by 2 expert Referees and I am pleased to tell you that it is considered to be acceptable for publication following satisfactory revision.

The reports are copied at the end of this email. Please address all of the points and incorporate all requested revisions, or explain in your Response to Referees why a change has not been made.

NEW POLICY: In order to improve the transparency of its peer review process The Journal of Physiology publishes online as supporting information the peer review history of all articles accepted for publication. Readers will have access to decision letters, including all Editors' comments and referee reports, for each version of the manuscript and any author responses to peer review comments. Referees can decide whether or not they wish to be named on the peer review history document.

Authors are asked to use The Journal's premium BioRender (<https://biorender.com/>) account to create/redraw their Abstract Figures. Information on how to access The Journal's premium BioRender account is here: <https://physoc.onlinelibrary.wiley.com/journal/14697793/biorender-access> and authors are expected to use this service. This will enable Authors to download high-resolution versions of their figures. The link provided should only be used for the purposes of this submission. Authors will be charged for figures created on this premium BioRender account if they are not related to this manuscript submission.

I hope you will find the comments helpful and have no difficulty returning your revisions within 4 weeks.

Your revised manuscript should be submitted online using the links in Author Tasks Link Not Available.

Any image files uploaded with the previous version are retained on the system. Please ensure you replace or remove all files that have been revised.

REVISION CHECKLIST:

- Article file, including any tables and figure legends, must be in an editable format (eg Word)
- Abstract figure file (see above)
- Statistical Summary Document
- Upload each figure as a separate high quality file
- Upload a full Response to Referees, including a response to any Senior and Reviewing Editor Comments;
- Upload a copy of the manuscript with the changes highlighted.

- A potential 'Cover Art' file for consideration as the Issue's cover image;
- Appropriate Supporting Information (Video, audio or data set https://jp.msubmit.net/cgi-bin/main.plex?form_type=display_requirements#supp).

To create your 'Response to Referees' copy all the reports, including any comments from the Senior and Reviewing Editors, into a Word, or similar, file and respond to each point in colour or CAPITALS and upload this when you submit your revision.

I look forward to receiving your revised submission.

If you have any queries please reply to this email and staff will be happy to assist.

Yours sincerely,

REQUIRED ITEMS:

-Papers must comply with the Statistics Policy https://jp.msubmit.net/cgi-bin/main.plex?form_type=display_requirements#statistics

In summary:

-If $n \leq 30$, all data points must be plotted in the figure in a way that reveals their range and distribution. A bar graph with data points overlaid, a box and whisker plot or a violin plot (preferably with data points included) are acceptable formats.

-If $n > 30$, then the entire raw dataset must be made available either as supporting information, or hosted on a not-for-profit repository e.g. FigShare, with access details provided in the manuscript.

-'n' clearly defined (e.g. x cells from y slices in z animals) in the Methods. Authors should be mindful of pseudoreplication.

-All relevant 'n' values must be clearly stated in the main text, figures and tables, and the Statistical Summary Document (required upon revision)

-The most appropriate summary statistic (e.g. mean or median and standard deviation) must be used. Standard Error of the Mean (SEM) alone is not permitted.

-Exact p values must be stated. Authors must not use 'greater than' or 'less than'. Exact p values must be stated to three significant figures even when 'no statistical significance' is claimed.

-Statistics Summary Document completed appropriately upon revision

EDITOR COMMENTS

Reviewing Editor:

The reviewers concur that the revision has been very careful and extensive. The revised manuscript presents a much stronger case for the comparison of the hand and face muscles in relation to local inhibitory and facilitatory circuits. Other concerns of the original version have equally been addressed appropriately. There remain some concerns related to the interpretation of results and therefore to conclusions, which are detailed in the reviewers' reports.

Senior Editor:

Comments for Authors to ensure the paper complies with the Statistics Policy (Required):
Please ensure that the presentation is in accordance with the Statistical Policy of the Journal. Specifically, raw data must be provided. In respect of the figures included in the present submission, this will entail that individual data points are plotted on the line graphs.

It may be necessary to alter the configuration of the figures in order to satisfy this requirement.

REFEREE COMMENTS

Referee #1:

The authors present have addressed all my previous comments. I believe their work has made for a more understandable and complete manuscript. I recommend publication.

Referee #2:

This is my second review of the research article, by Ginatempo and colleagues, with a new title, "Is it possible to compare inhibitory and excitatory intracortical circuits in face and hand primary motor cortex?" I appreciate the authors carefully addressed my previous concerns, especially the rationale of directly comparing the two muscles. The quality of the

manuscript has been greatly improved. The authors present a strong case in that we can compare the inhibitory circuitry and properties of the hand and face muscles. This research may serve as a starting point for studies comparing hand and face motor functions and their interactions. I have the following comments for the authors to address before the paper can be published.

My major comment regarding the interpretation of the results is that it is puzzling why would the two muscles show opposite trends for resting and active SICI. While the Discussion largely focuses on the less resting SICI in DAO. I was wondering what would account for the higher active SICI in DAO? If it's due to a baseline effect, it should affect resting and active SICI similarly.

Minor comments:

Line 425 on page 15 states that "the size and intrinsic properties of the cortical neurons are the same in the hand motor cortex." This is a strong assumption. Please cite references. To my knowledge, there are subtypes of pyramidal neurons (e.g., Kasper et al., 1994). It is also not clear how relevant this statement is to the rest of the argument. If DAO is supported by the corticobulbar tract whereas FDI via corticospinal, why would we assume the cortical neurons are sending input to these two different pathways need to have the same properties?

In the Key Points summary, the second point states that "At rest, intracortical inhibitory activity was stronger in DAO than FDI." This is probably a typo. The results showed the opposite trend.

CSP was first mentioned in the last paragraph of the Introduction. It is not clear what is the motivation for using this protocol. There needs to be a brief but clear introduction and justification of this stimulation protocol.

In the Results section, sometimes "MEP inhibition" was used to refer to SICI, and "MEP facilitation" was used to refer to SICF (e.g., the first sentences in Experiment 1, 3, and 4 results). Are these expressions interchangeable?

Please be consistent.

Similarly, "effective" was used in many places to describe SICI and SICF. It's a bit vague what "effective" means. Please define it early on in the paper.

In the first paragraph of the Discussion, the last sentence states, "SICF was similar in magnitude in DAO and DFI...", but SICF is a ratio, not MEP magnitude. Also, this should be "resting SICF," correct?

Some of the significant results were shown with a star in the figures, while some were not. Please double-check the results and make them consistent throughout Fig. 1-7.

END OF COMMENTS

1st Confidential Review

23-Mar-2022

Reviewer Response to Manuscript JP-RP-2022-283137X

The authors present have addressed all my previous comments. I believe their work has made for a more understandable and complete manuscript. I recommend publication.

Manuscript Number: **JP-RP-2022-283137X**

EDITOR COMMENTS

Reviewing Editor:

The reviewers concur that the revision has been very careful and extensive. The revised manuscript presents a much stronger case for the comparison of the hand and face muscles in relation to local inhibitory and facilitatory circuits. Other concerns of the original version have equally been addressed appropriately. There remain some concerns related to the interpretation of results and therefore to conclusions, which are detailed in the reviewers' reports.

We thank the Editor for giving us the opportunity to revise the Manuscript.

Senior Editor:

Comments for Authors to ensure the paper complies with the Statistics Policy (Required): Please ensure that the presentation is in accordance with the Statistical Policy of the Journal. Specifically, raw data must be provided. In respect of the figures included in the present submission, this will entail that individual data points are plotted on the line graphs.

It may be necessary to alter the configuration of the figures in order to satisfy this requirement.

In order to ensure the paper to comply with the Statistics Policy of Journal of Physiology the figures have been changed accordingly.

REFEREE COMMENTS

Referee #1:

The authors present have addressed all my previous comments. I believe their work has made for a more understandable and complete manuscript. I recommend publication.

We thank the Reviewer for his/her words of appreciation of our study.

Referee #2:

This is my second review of the research article, by Ginatempo and colleagues, with a new title, "Is it possible to compare inhibitory and excitatory intracortical circuits in face and hand primary motor cortex?" I appreciate the authors carefully addressed my previous concerns, especially the rationale of directly comparing the two muscles. The quality of the manuscript has been greatly improved. The authors present a strong case in that we can compare the inhibitory circuitry and properties of the hand and face muscles. This research may serve as a starting point for studies

comparing hand and face motor functions and their interactions. I have the following comments for the authors to address before the paper can be published.

My major comment regarding the interpretation of the results is that it is puzzling why would the two muscles show opposite trends for resting and active SICI. While the Discussion largely focuses on the less resting SICI in DAO. I was wondering what would account for the higher active SICI in DAO? If it's due to a baseline effect, it should affect resting and active SICI similarly.

We apologize for not discussing appropriately the possible mechanisms and functional significance of the stronger SICI in DAO than FDI in the active condition. It seems unreasonable that the difference in SICI between DAO and FDI in the active condition depends on the amplitude of the baseline MEP, since if this was the case the DAO-FDI difference in SICI, would have been the same in resting and active conditions. Possible reasons for this difference have been widely discussed in the revised manuscript (lines: 507-527).

Minor comments:

Line 425 on page 15 states that "the size and intrinsic properties of the cortical neurons are the same in the hand motor cortex." This is a strong assumption. Please cite references. To my knowledge, there are subtypes of pyramidal neurons (e.g., Kasper et al., 1994). It is also not clear how relevant this statement is to the rest of the argument. If DAO is supported by the corticobulbar tract whereas FDI via corticospinal, why would we assume the cortical neurons are sending input to these two different pathways need to have the same properties?

We see the Reviewer point. We did not mean to state that the size and intrinsic properties of the cortical neurons are the same in the hand and face motor cortex. However, we may assume this as true, if we consider that we are dealing with the primary motor cortex, where, as far as we know, no differences have been described among pyramidal neurons of the different muscle representation areas. This assumption allows hypothesizing that the corticomotor output to the lower motor neurons is the same for both muscles and then it should not be considered as a confounding factor.

In the Key Points summary, the second point states that "At rest, intracortical inhibitory activity was stronger in DAO than FDI." This is probably a typo. The results showed the opposite trend.

The Reviewer is right, it was a typo error. The sentence has been amended (line:38)

CSP was first mentioned in the last paragraph of the Introduction. It is not clear what is the motivation for using this protocol. There needs to be a brief but clear introduction and justification of this stimulation protocol.

We are sorry for being unclear on this important issue. In the revised manuscript the rational and a brief introduction of CSP protocol have been added (lines: 92-97, 115-116).

In the Results section, sometimes "MEP inhibition" was used to refer to SICI, and "MEP facilitation"

was used to refer to SICF (e.g., the first sentences in Experiment 1, 3, and 4 results). Are these expressions interchangeable? Please be consistent.

We apologize for being not consistent when referring to SICI and SICF. Now amended.

Similarly, "effective" was used in many places to describe SICI and SICF. It's a bit vague what "effective" means. Please define it early on in the paper.

The manuscript has been revised specifying what effective means.

In the first paragraph of the Discussion, the last sentence states, "SICF was similar in magnitude in DAO and DFI...," but SICF is a ratio, not MEP magnitude. Also, this should be "resting SICF," correct?

The reviewer is right. The sentence has been changed accordingly (lines: 396-397).

Some of the significant results were shown with a star in the figures, while some were not. Please double-check the results and make them consistent throughout Fig. 1-7.

Done.

Dear Dr Deriu,

Re: JP-RP-2022-283137XR1 "Is it possible to compare inhibitory and excitatory intracortical circuits in face and hand primary motor cortex?" by Francesca Ginatempo, Nicola Loi, Andrea Manca, John C Rothwell, and Franca Deriu

I am pleased to tell you that your paper has been accepted for publication in The Journal of Physiology.

NEW POLICY: In order to improve the transparency of its peer review process The Journal of Physiology publishes online as supporting information the peer review history of all articles accepted for publication. Readers will have access to decision letters, including all Editors' comments and referee reports, for each version of the manuscript and any author responses to peer review comments. Referees can decide whether or not they wish to be named on the peer review history document.

The last Word version of the paper submitted will be used by the Production Editors to prepare your proof. When this is ready you will receive an email containing a link to Wiley's Online Proofing System. The proof should be checked and corrected as quickly as possible.

Authors should note that it is too late at this point to offer corrections prior to proofing. The accepted version will be published online, ahead of the copy edited and typeset version being made available. Major corrections at proof stage, such as changes to figures, will be referred to the Reviewing Editor for approval before they can be incorporated. Only minor changes, such as to style and consistency, should be made a proof stage. Changes that need to be made after proof stage will usually require a formal correction notice.

All queries at proof stage should be sent to TJP@wiley.com

Are you on Twitter? Once your paper is online, why not share your achievement with your followers. Please tag The Journal (@jphysiol) in any tweets and we will share your accepted paper with our 23,000+ followers!

Yours sincerely,

Richard Carson
Senior Editor
The Journal of Physiology

P.S. - You can help your research get the attention it deserves! Check out Wiley's free Promotion Guide for best-practice recommendations for promoting your work at www.wileyauthors.com/eeo/guide. And learn more about Wiley Editing Services which offers professional video, design, and writing services to create shareable video abstracts, infographics, conference posters, lay summaries, and research news stories for your research at www.wileyauthors.com/eeo/promotion.

*** IMPORTANT NOTICE ABOUT OPEN ACCESS ***

To assist authors whose funding agencies mandate public access to published research findings sooner than 12 months after publication The Journal of Physiology allows authors to pay an open access (OA) fee to have their papers made freely available immediately on publication.

You will receive an email from Wiley with details on how to register or log-in to Wiley Authors Services where you will be able to place an OnlineOpen order.

You can check if your funder or institution has a Wiley Open Access Account here <https://authorservices.wiley.com/author-resources/Journal-Authors/licensing-and-open-access/open-access/author-compliance-tool.html>

Your article will be made Open Access upon publication, or as soon as payment is received.

If you wish to put your paper on an OA website such as PMC or UKPMC or your institutional repository within 12 months of publication you must pay the open access fee, which covers the cost of publication.

OnlineOpen articles are deposited in PubMed Central (PMC) and PMC mirror sites. Authors of OnlineOpen articles are permitted to post the final, published PDF of their article on a website, institutional repository, or other free public server, immediately on publication.

Note to NIH-funded authors: The Journal of Physiology is published on PMC 12 months after publication, NIH-funded authors DO NOT NEED to pay to publish and DO NOT NEED to post their accepted papers on PMC.

EDITOR COMMENTS

Thank you for having carefully addressed all comments.